# Graph Harmony: Denoising and Nuclear-Norm Wasserstein Adaptation for Enhanced Domain Transfer in Graph-Structured Data

## Abstract

Graph-structured data can be found in numerous domains, yet the scarcity of labeled instances hinders its effective utilization of deep learning in many scenarios. Traditional unsupervised domain adaptation (UDA) strategies for graphs primarily hinge on adversarial learning and pseudo-labeling. These approaches fail to effectively leverage graph discriminative features, leading to class mismatching and unreliable label quality. To address these obstacles, we develop the Denoising and Nuclear-Norm Wasserstein Adaptation Network (DNAN). DNAN employs the Nuclear-norm Wasserstein discrepancy (NWD), which can simultaneously achieve domain alignment and class distinguishment. It also integrates a denoising mechanism via a Variational Graph Autoencoder. This denoising mechanism helps capture essential features of both source and target domains, improving the robustness of the domain adaptation process. Our comprehensive experiments demonstrate that DNAN outperforms state-of-the-art methods on standard UDA benchmarks for graph classification.

## 1 Introduction

While deep learning has made substantial progress in handling graph-structured data, it shares a drawback with other methods in the same category — a heavy reliance on labeled data. This requirement presents a significant obstacle in real-world applications, where the gathering and annotating of graph-structured data come with a steep price tag, both in terms of time and resources. Obtaining detailed labels for graph-structured data, e.g., chemical molecules, is a considerable challenge because chemical molecules are incredibly complex, comprising a large number of atoms connected in various ways through different kinds of bonds. Collecting annotated graph-structured data like social networks is also challenging due to the need to protect personal and sensitive information and the continual changes in network relationships. The label scarcity makes it difficult to derive meaningful insights and hinders the development of strategies and solutions based on deep learning. Therefore, it is highly desirable to relax the need for extensive graph-structured data annotation to replicate the success of deep learning in applications.

To navigate the challenge of label scarcity, Unsupervised Domain Adaptation (UDA) (Ganin & Lempitsky, 2015) has emerged as a promising frontier, with the aim of leveraging labeled data from a related source domain to inform an unlabeled target domain. The challenge of label scarcity requires Unsupervised Domain Adaptation (UDA) due to the absence of labels in the target domain. The target domain is our primary area of interest. To overcome this challenge, we utilize a source dataset rich in labels for training our model. However, the inherent difference between the source and target datasets requires the application of domain adaptation strategies. These strategies enable the effective application of models trained on the well-labeled source data to achieve high performance on the target domain, despite its lack of labels. The principle of UDA is to align the data distributions between the two domains within a common embedding space, enabling a classifier trained on the source domain to perform competently on the target domain.

While UDA has been extensively applied to array-structured data (Long et al., 2016; Kang et al., 2019), its translation to graph-structured data remains under-explored. Graph samples have a wide range of structural

variations, including differences in connectivity patterns, node degrees, and subgraph structures. The primary challenge in applying UDA to graph classification is the domain shift in structural patterns, or simply, the structural variations between the source and target domain graphs. These structural variations make it challenging for models to identify and leverage invariant features across domains. Pioneering methods, such as DANE (Zhang et al., 2019a), integrate generative adversarial networks (GANs) with graph convolutional networks (GCNs) to align the domains. Others, like the approach by Wu et al. (2020), introduce attention mechanisms to reconcile global and local consistencies, again employing GANs for cross-domain node embedding extraction. However, these GAN-based methods have the drawback of class mismatching, lacking clear separability between features from different classes, as they align target and source domain features irrespective of their classes. In addition, these methods are designed for node classification. The UDA strategy for graph classification has not been well-explored.

This paper focuses on the UDA setting for graph classification. We propose the Denoising and Nuclear-Norm Wasserstein Adaptation Network (DNAN) to address the primary challenges in graph UDA tasks and problems in the previous GAN-based methods. Our DNAN benefits from the denoising mechanism with a Variational Graph Autoencoder (VGAE) and the Nuclear-Norm Wasserstein Discrepancy. By leveraging the Nuclear-Norm Wasserstein Discrepancy, it tackles the class mismatch issue in existing graph-based UDA methods. Unlike the previous GAN-based methods, DNAN performs a refined, class-specific alignment of source and target domain distributions within a shared embedding space, preserving the distinct separability of features across classes. The inclusion of the denoising mechanism is motivated by the stuctural variations between source and target domain graphs for graph UDA setting. The denoising mechanism of VGAE reconstructs clean adjacency matrices from corrupted versions. This process forces the model to learn robust features that are more invariant to structural variations and helps the model focus on the underlying structure and features that are relevant to the classification task. Thus, we believe the denoising mechanism could help handle the domain shift in UDA tasks for graph classification. By using these two components, DNAN performs competitively and achieves state-of-the-art performance on major UDA benchmarks for graph classification.

Our contributions mainly lie in applying existing techniques to a new problem and introducing a new effective combination of existing approaches. Our first contribution is applying denoising techniques to address the domain shift in structural patterns in the graph UDA problem. This utilization of the denoising mechanism is not trivial, and we believe we're the first to apply the denoising mechanism in the graph UDA context. Our second contribution is integrating Nuclear-norm Wasserstein Discrepancy (NWD) with VGAE. Typically, VGAEs are used together with a domain classifier in previous domain adaptation methods. We use NWD to remove the domain classifier and incorporate the domain adaptation directly into our class classifier. This integration hasn't been done before.

## 2 Related Work

**Unsupervised Domain Adaptation** A foundational approach within UDA is to reduce the discrepancy between the source and target domain distributions using adversarial learning. A representative method in this space, the Domain Adversarial Neural Network (DANN) (Ganin & Lempitsky, 2015), employs an adversarial training framework to align domain representations by confusing a domain classifier in a shared embedding space. This strategy is adapted from generative adversarial networks (GANs) (Goodfellow et al., 2020), tailored for domain adaptation purposes. Expanding on this adversarial methodology, the FGDA technique (Gao et al., 2021) used a discriminator to discern the gradient distribution of features, thereby achieving better performance in mitigating domain discrepancy. Furthermore, DADA (Tang & Jia, 2020) proposed an innovative strategy by integrating the domain-specific classifier with the domain discriminator to align the joint distributions of two domains more effectively.

Adversarial approaches are complemented by statistical discrepancy measures like Maximum Mean Discrepancy (MMD), utilized in the Joint Distribution Optimal Transport (JDOT) model (Courty et al., 2017b). WD has been leveraged for distribution alignment in UDA methods (Courty et al., 2017a; Damodaran et al., 2018), with Redko et al. (2017) providing theoretical foundations for model generalization on the target domain when employing WD. However, the practical application of WD is computationally intensive due to

the absence of a closed-form solution. The Sliced Wasserstein Distance (SWD) (Rabin et al., 2011; Bonneel et al., 2015) offers a computationally feasible alternative. Reconstruction-based objectives constitute another research direction, enforcing feature invariance across domains by reconstructing source domain data from target domain features, as in the work by Ghifary et al. (2016). Additionally, the application of Variational Autoencoders (VAEs) Kingma & Welling (2013) to UDA, such as in the Variational Fair Autoencoder (Louizos et al., 2015), showcases the capabilities of probabilistic generative models in domain-invariant feature learning. Our proposed method draws inspiration from the Variational Autoencoder's framework. Other notable approaches like ToAlign (Wei et al., 2021), SDAT (Rangwani et al., 2022), and BIWAA (Westfechtel et al., 2023), mark the recent advancement in UDA, surpassing previous models in performance. These three approaches are detailed in the experiment sections as our references for current state-of-the-art methods. However, extending these existing methods to graph-structured data is often non-trivial.

**Graph Representation Learning** Graph representation learning (GRL) has emerged as an important approach in machine learning, tasked with distilling complex graph-structured data into a tractable, low-dimensional vector space to enable the use of architectures developed for array-structured data. Previously, spectral methods laid the foundation, leveraging graph Laplacians to capture topological structures of graphs despite the limitations in scalability for larger graphs (Belkin & Niyogi, 2003; Chung, 1997). The field then evolved with algorithms such as DeepWalk (Perozzi et al., 2014) and Node2Vec (Grover & Leskovec, 2016), which utilized random walks to encode local neighborhood structures into node embeddings, balancing the preservation of local and global graph characteristics. The introduction of Graph Neural Networks (GNNs) marked a significant advancement in GRL. GNNs, specifically Graph Convolutional Networks (GCNs), offer a way to generalize neural network approaches to graph data, integrating neighborhood information into node embeddings (Kipf & Welling, 2016a). This was further refined by GraphSAGE, which scaled GNNs by learning a function to sample and aggregate local neighborhood features (Hamilton et al., 2017b;a). Moreover, Graph Attention Networks (GATs) introduced an attention mechanism, enabling the model to adaptively prioritize information from different parts of a node's neighborhood, thus enhancing the expressiveness of the embeddings (Velickovic et al., 2017). These advances, along with the development of graph autoencoders like VGAEs that focused on graph reconstruction from embeddings, have broadened the applications of GRL and continue to shape its trajectory (Kipf & Welling, 2016b). For a fair comparison, when we compare with methods originally not proposed for graph domain adaptation, we replace their feature extraction backbones with GAT. This is because GAT is the graph encoder we use in our approach.

## 3  Problem Description

We operate under the assumption that there is a source domain with labeled data and a target domain with unlabeled data. In both domains, each input instance is a graph-structured data sample. Our primary objective is to develop a predictive model for the target domain by transferring knowledge from the source domain.

**Graph Classification** We focus on a graph classification task, where a graph sample can be represented as $G = (X, A)$. $X \in \mathbb{R}^{n \times K}$, where $n$ represents the number of nodes in $G$ and $K$ represents the dimension of the features for each node. Note that the number of nodes may vary across different graphs. Specifically, $x_i \in X$ corresponds to the feature associated with a node $v_i$. Let $A \in \mathbb{R}^{n \times n}$ denote the adjacency matrix. The matrix $A$ encapsulates the topological structure of $G$. Each graph is associated with a class, and we use $y$ to denote the ground truth label of the graph sample $G$. The goal is to train a model capable of classifying graphs effectively and accurately.

**Source Domain Dataset** We consider a fully labeled source domain dataset as $(D_s, Y_s) = (\{G_s^k\}, \{y_s^k\})$, where $G_s^k$ is the $k^{th}$ batch of graph samples in $D_s$ and $y_s^k$ is the ground truth labels of $G_s^k$.

**Target Domain Dataset** We consider that only an unlabeled target domain dataset $D_t = \{G_t^k\}$ is accessible, where $G_t^k$ is the $k^{th}$ batch of graph samples in the target dataset $D_t$.

**UDA for Graph Classification** The comprehensive pipeline of an UDA model for graph classification begins by leveraging a neural network to extract relevant features from the input graph samples. Then, it aligns the domains by utilizing either a distance metric to minimize the discrepancy between the source and

target feature distributions or employs adversarial techniques to achieve domain-invariant feature representations. Subsequently, the classification task is performed using the aligned feature. Usually, the classifier for the classification task is trained with source labels, but different UDA models may have different strategies on training the classifier. The performance of the UDA model is evaluated based on how well it performs the classification task on the test portion of the target domain dataset.

## 4 Proposed Method

Figure 1 visualizes a high-level description of our proposed pipeline. Our algorithm benefits from a denoising mechanism via Variational Graph Autoencoder (VGAE) and the Nuclear-norm Wasserstein discrepancy for distribution alignment. In a nutshell, our method embeds the graph-structured data from both domains into a shared feature space via VGAE with a denoising mechanism. Then, we align the distributions of both domains in this shared feature space by minimizing the Nuclear-norm Wasserstein discrepancy.

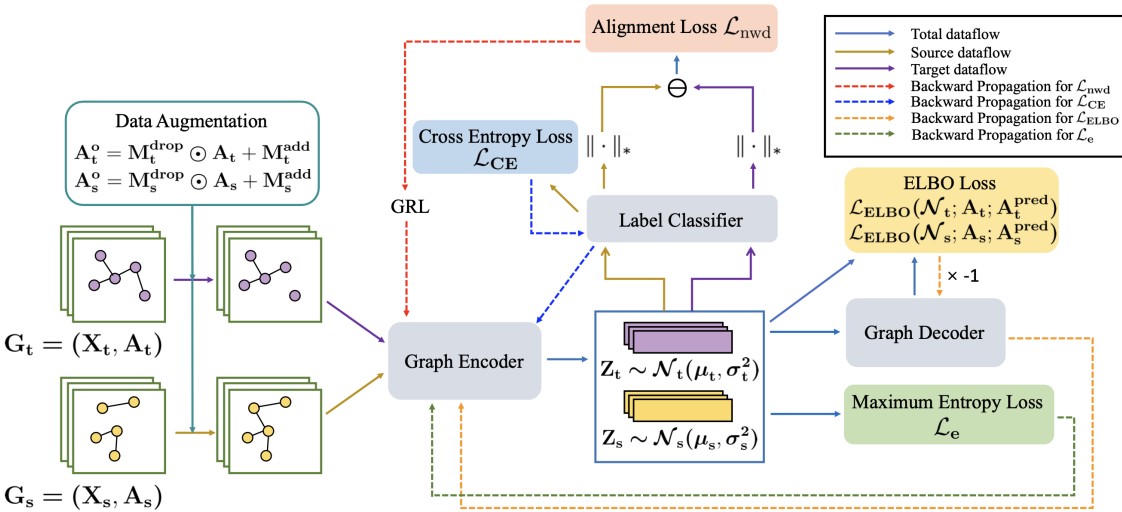

Figure 1: The block-diagram visualization of DNAN: The inputs are source batch $G_s$ and target batch $G_t$. We first add noise to the graph samples of the source and target batches by applying data augmentation to their adjacency matrices $A_s$ and $A_t$, using masks $M_t^{\mathrm{drop}}$, $M_s^{\mathrm{drop}}$, $M_s^{\mathrm{add}}$, and $M_t^{\mathrm{add}}$. Then, the graph encoder of the VGAE produces the latent variables $Z_s$ and $Z_t$ from node features $X_s, X_t$ and augmented adjacency matrices $A_s^o, A_t^o$. We train a label classifier using a cross-entropy loss $\mathcal{L}_{\mathrm{CE}}$ between the output of the label classifier and the ground-truth label. To align the latent variables of both domains, we compute a Nuclear-norm Wasserstein discrepancy (NWD) using $Z_s$, $Z_t$, and the label classifier. The graph decoder of VGAE reconstructs the original adjacency matrices $A_s, A_t$ from $Z_s, Z_t$. Then, the Evidence Lower Bound (ELBO) loss is computed based on the outputs of the graph encoder and original and reconstructed adjacency matrices. Lastly, the model applies maximum entropy regularization $\mathcal{L}_e$ to the latent variables $Z_t, Z_s$.

### 4.1 Latent Variables Construction with Denoising Mechanism

In a system composed of the random variables $X$ and $Z$, $X$ denotes the observed variable, while Z is the latent variable. The conditional probability density $P(X|Z)$ is known as the likelihood. From Bayes' theorem, we can compute the posterior probability density $P(Z|X)$ as

$$P(Z|X) = \frac{P(X|Z)P(Z)}{P(X)} \tag{1}$$

The marginal probability density $P(X)$ can be computed as

$$P(X) = \int_Z P(X|Z)P(Z)\, dZ \tag{2}$$

The marginal probability density $P(X)$ is known as the evidence and $p(Z)$ is termed the prior probability density, as it encapsulates the prior information regarding $Z$.

Variational inference (Blei et al., 2017) is an estimation technique aimed at approximating the complex, often intractable posterior distribution $P(Z|X)$ with a more computationally manageable parameterized distribution $q_\phi(Z)$. A Variational Autoencoder (VAE) is a type of generative model that uses variational inference within a probabilistic framework to encode input data into a latent space and reconstruct outputs from this space. This process allows for the generation of new data points that are similar to the original inputs. In a VAE, the model defines an approximate distribution, $q_\phi(Z|X)$, which is conditioned on an input observation $X$. This distribution typically takes the form of a neural network, where $X$ serves as the input and the latent variable $Z$ as the output. The parameters $\phi$, representing the neural network's weights, are shared across all input observations $X$. This neural network, known as the inference network, essentially learns to encode the input data into a latent representation. Conversely, the model also defines a parameterized distribution $P_\theta(X|Z)$ that models the probability of observing $X$ given the latent variable $Z$. $P_\theta(X|Z)$ is also typically chosen as a neural network with $Z$ as the input and a distribution over possible values of $X$ as the output. The weights of this neural network are denoted by $\theta$, and this network is referred to as the generative network. The generative network is tasked with decoding latent representations back into data points, thereby enabling the generation of new data points by sampling from the latent space. The prior distribution for the latent variable $Z$, denoted as $P(Z)$, is typically chosen to be an isotropic Gaussian distribution, expressed as $\mathcal{N}(0, \sigma I)$. To train a VAE, we maiximize the following lower bound, known as the evidence lower bound (ELBO), with regard to the parameters $\theta$ and $\phi$.

$$\mathcal{L}_{\text{ELBO}} = \mathbb{E}_{q_\phi(Z|X)} \left[ \frac{\log P_\theta(X, Z)}{q_\phi(Z|X)} \right] = \mathbb{E}_{q_\phi(Z|X)} \left[ \log P_\theta(X|Z) \right] - \text{KL} \left( q_\phi(Z|X) \, || \, P(Z) \right) \tag{3}$$

where KL represents the Kullback-Leibler divergence.

The Variational Graph Autoencoders (Kipf & Welling, 2016b) is based on Variational Autoencoder concepts and is especially for graph-structured data. Given a graph sample $G = (X, A)$ with $n$ nodes, the graph encoder (inference network) in VGAE generates a corresponding latent variable $Z$. $q_\phi(Z|A, X)$ is used to denote the graph encoder, characterized by the parameter $\phi$. $q_\phi(Z|A, X)$ aims to approximate the real posterior distribution $P(Z|A, X)$. The graph decoder (generative network) of a standard VGAE is represented as $P_\theta(A|Z)$, defined by parameters $\theta$. The prior distribution is denoted by $P(Z)$, assumed to be a normal distribution, specifically $P(Z) \sim \mathcal{N}(0, I)$. The ELBO for standard VGAE is given as:

$$\mathcal{L}_{\text{ELBO}} = E_{q_\phi(Z|A,X)}[\log P_\theta(A|Z)] - \text{KL}(q_\phi(Z|A, X) || P(Z)) \tag{4}$$

The standard VGAE is trained through maximizing the $\mathcal{L}_{\text{ELBO}}$.

Instead using plain VGAE, we proposed to use VGAE with denoising mechanism. Particularly, we adopt the denoising criterion of Denoising Variational Autoencoders (DVAE) (Im Im et al., 2017) and translate it to varitaionl graph autoencoder. Like DVAE, the VGAE with denoising mechanism reconstructs clean graph data from inputs perturbed with noise. Similar to the training process of VGAE, however, we have some variations. Given a graph sample $G = (X, A)$, we train the VGAE on both $G = (X, A)$ and $G^o = (X, A^o)$, where $A^o$ is the adjacency matrix with noise. We implement data augmentation to add noise to the adjacency matrices. Specifically, we benefit from a random manipulation-based approach (Cai et al., 2021). To this end, edges are dropped and added randomly by modifying the values in the adjacency matrix $A$ of the original graph. $A^o$ is constructed as follows:

$$A^o = M^{\text{drop}} \odot A + M^{\text{add}}, \quad m_{ij}^{\text{add}} \sim \text{Bernoulli}(p^{\text{add}} \cdot p^{\text{edge}}), \quad m_{ij}^{\text{drop}} \sim \text{Bernoulli}(p^{\text{drop}}) \tag{5}$$

where $p^{\text{add}}$, $p^{\text{edge}}$, and $p^{\text{drop}}$ denote the edge addition rate, the sparsity of the adjacency matrix $A$, and the edge dropping rate. $\odot$ represents the element-wise multiplication between two matrices. $M^{\text{drop}}$ and $M^{\text{add}}$ represent mask matrices with the same dimensions as $A$. For each element $m_{ij}^{\text{add}} \in M^{\text{add}}$ or $m_{ij}^{\text{drop}} \in M^{\text{drop}}$, we sample its value from a Bernoulli distribution.

When the input is $A$, the evidence lower bound for VGAE with denoising mechanism is same as Equation 4. When the input is $A^o$, we modify the evidence lower bound and called the modified variational lower bound

to be $\mathcal{L}_{\text{dvgae}}$. $\mathcal{L}_{\text{dvgae}}$ is presented as follows.

$$\mathcal{L}_{\text{dvgae}} = \mathbb{E}_{\tilde{q}_\phi(Z|A,X)} \left[ \log \frac{P_\theta(A,Z)}{q_\phi(Z|A^o,X)} \right] \tag{6}$$

The network $\tilde{q}_\phi(Z|A,X)$ is defined to include the component $q_\phi(Z|A^o,X)$ along with an additional stochastic layer. This stochastic layer takes the adjacency matrix $A$ as its input and produces the distribution of noised-perturbed $A$ ($A^o$) as its output. In section B of Appendix, a comprehensive description of $\tilde{q}_\phi(Z|A,X)$ and the reason to use an extra stochastic layer is provided. We also show $\mathcal{L}_{\text{dvgae}}$ is a valid variational lower bound. Specifically, we prove the following theorem in Appendix.

**Theorem 1.** Maximizing $\mathcal{L}_{\text{dvgae}}$ is equivalent to minimizing the following objective

$$\mathbb{E}_{P(A^o|A)} \left[ \text{KL} \left( q_\phi(Z|A^o,X) || P(Z|A,X) \right) \right] \tag{7}$$

In other words,

$$\log P_\theta(A) = \mathcal{L}_{\text{dvgae}} + \mathbb{E}_{P(A^o|A)} \left[ \text{KL} \left( q_\phi(Z|A^o,X) || P(Z|A,X) \right) \right]$$

Though $\mathcal{L}_{\text{dvgae}}$ is different from $\mathcal{L}_{\text{ELBO}}$ and can not be directly computed, we apply the Monte Carlo sampling to approximate $\mathcal{L}_{\text{dvgae}}$.

$$\mathcal{L}_{\text{dvgae}} \approx \frac{1}{M} \sum_{m=1}^{M} \mathbb{E}_{q_\phi(Z|A_m^o,X)} \left[ \log P_\theta(A|Z) \right] - \text{KL}(q_\phi(Z|A_m^o,X) || P(Z)) \tag{8}$$

The derivation of this approximation is also in section B of the Appendix. With this approximation, the training procedure turns to be similar to how the regular VAE is trained except that the input is corrupted by a noise distribution.

We utilize Graph Attention Networks (GAT) as the graph encoder $q_\phi$ of the VGAE. The encoding equations when the input is $A^o$ are given as follows:

$$\begin{aligned}
\mu &= \text{GAT}_\mu(A^o, X) \\
\log \sigma &= \text{GAT}_\sigma(A^o, X) \\
z_i &= \mu_i + \varepsilon_i \cdot \sigma_i, \ \varepsilon_i \sim \mathcal{N}(0,1) \\
q_\phi(z_i|A^o, X) &= \mathcal{N}(z_i|\mu_i, \text{diag}(\sigma_i^2)) \\
q_\phi(Z|A^o, X) &= \prod_{i=1}^{n} q_\phi(z_i|A^o, X)
\end{aligned} \tag{9}$$

The element $z_i$ corresponds to the $i^{th}$ row of $Z$. This same row-wise correspondence applies to $\mu_i$ and $\log \sigma_i$ as well. Using the reparameterization trick, we transform the generated $\mu_i$ and $\sigma_i$ into latent variable $z_i$. To achieve a cleaner construction of the latent variable, we apply an element-wise maximum entropy loss, $\mathcal{L}_e$, as a regularization term. This maximum entropy loss removes irrelevant information from the latent variable, enhancing its clarity and effectiveness. The specifics of the maximum entropy loss are described below:

$$\begin{aligned}
\mathcal{L}_{\text{e}} &= \frac{1}{N_s + N_t} \sum_{G^k \in (D_s, D_t)} \text{ME}(Z^k) \\
\text{ME}(Z^k) &= \frac{1}{n_k \times F} \sum_{i=1}^{n_k} \sum_{j=1}^{F} \sigma(z_{ij}) \log \sigma(z_{ij})
\end{aligned} \tag{10}$$

where $n_k$ is the number of nodes in the graph sample $G^k$ and $F$ denotes the dimension of $G^k$'s latent variable $Z^k$. After obtaining the latent variable $Z$, an inner product decoder $P_\theta(A|Z)$ is applied to $Z$ to reconstruct the adjacency matrix before data augmentation. This decoder translates each pair of node representations into a binary value, indicating whether an edge exists in the reconstructed adjacency matrix $A^o$. Specifically,

we first use an MLP (multilayer perceptron) described by parameters $\{W_0, W_1\}$ to improve the expressive capacity of the latent variable $Z$. Then, we compute the dot product for each node representation pair as:

$$H = \text{ReLU}((Z \cdot W_0) \cdot W_1)$$
$$p(A_{ij}^o = 1|h_i, h_j) = \sigma(h_i^T h_j)$$
$$p(A^o|Z) = \prod_{i=1}^{n}\prod_{j=1}^{n} p(A_{ij}^o|h_i, h_j) \tag{11}$$

where $h_i$ represents the $i^{th}$ row of $H$ and $A_{ij}^o$ is an element of reconstructed adjacency matrix $A^o$. The parameter $\theta$ describes the graph decoder includes $\{W_0, W_1\}$.

### 4.2 Distribution Alignment

By using $\mathcal{L}_{\text{dvgae}}$, the graph encoder of VGAE is better equipped to grasp the essential features. However, we still face a crucial challenge: addressing the performance degradation that occurs when a model trained on data from a source domain is applied to a target domain with a different data distribution. As mentioned in the previous sections, traditional approaches in unsupervised domain adaptation often use a domain discriminator that engages in a min-max game with a feature extractor to produce domain-invariant features. However, these methods primarily focus on confusing features at the domain level, which might negatively impact class-level information and lead to the mode collapse problem (Kurmi & Namboodiri, 2019; Tang & Jia, 2020). To address these challenges, our approach integrates the Nuclear-norm Wasserstein discrepancy (NWD) (Chen et al., 2022) to effectively align the source and target domains' feature representations while maintaining class-level discrimination by considering it as a loss function. In section A of the Appendix, we provide the detailed background of NWD for reference. The NWD addresses the class mismatch issue by incorporating class information into the domain adaptation process. The class classifier not only performs class classification but also serves as a domain discriminator. The class classifier is capable of identifying correlations both within and among different classes. These correlations, however, vary between the source domain data and the target domain data. To achieve domain alignment, NWD aligns the correlations within and between classes in the target domain with those in the source domain. This alignment ensures consistency of class across different domains, so the class mismatch problem is mitigated. .

Our method utilizes a Variational Graph Autoencoder described in the previous section and a classifier $C$ with parameter $\theta_c$. We construct $C$ with two fully connected layers. The empirical NWD loss is defined as:

$$\mathcal{L}_{\text{nwd}} = \frac{1}{N_s^{\text{train}}}\sum_{k=1}^{N_s^{\text{tran}}}\|C(Z_s^k)\|_* - \frac{1}{N_t^{\text{train}}}\sum_{k=1}^{N_t^{\text{train}}}\|C(Z_t^k)\|_* \tag{12}$$

where $Z_s^k$ represents the latent variables for the $k$-th batch of graph samples $G_s^k$ and where $Z_t^k$ represents the latent variables for the $k$-th batch of graph samples $G_t^k$. $\|\cdot\|_*$ denotes the Nuclear norm. $N_s^{\text{train}}$ is the number of training batches in the source dataset, and $N_t^{\text{train}}$ is the number of training batches in the target dataset. To avoid complex alternating updates, we employ a Gradient Reverse Layer (GRL) (Ganin et al., 2016), which allows for updating in a single backpropagation step. The distribution alignment is achieved through a min-max game, optimized as:

$$\min_{\phi}\max_{\theta_c} \mathcal{L}_{\text{nwd}} \tag{13}$$

### 4.3 Algorithm Summary

In addition to distribution alignment, to ensure accurate classification, we optimize the graph encoder in VGAE and the classifier $C$ using a supervised classification loss $\mathcal{L}_{\text{cls}}$ for the source domain:

$$\mathcal{L}_{\text{cls}} = \frac{1}{N_s^{\text{train}}}\sum_{j=1}^{N_s^{\text{train}}} \mathcal{L}_{\text{CE}}(C(Z_s^k, y_s^k)) \tag{14}$$

---

**Algorithm 1** DNAN Method

---

**Input**: $(D_s, Y_s), D_t$
**Parameters**: VGAE parameters $\{\phi$ (Graph Encoder), $\theta$ (Graph Decoder)$\}$, Classifier parameter $\{\theta_c\}$
**Output**: Trained Parameters $\phi, \theta, \theta_c$

1: Randomly sample a batch of $\{(G_s^k, y_s^k)\}$
2: Randomly sample a batch of $\{G_k^t\}$
3: Forward Propagation
4: Update $\phi, \theta, \theta_c$ based on Equation (10)
5: Add noise to $\{G_s^k\}, \{G_t^k\}$ based on Equation (2)
6: Forward Propagation
7: Update $\phi, \theta, \theta_c$ based on Equation (10)
8: **return** $\phi, \theta, \theta_c$

---

Then, by combining all the loss described in the previous sections, our total optimization object is formulated as follows:

$$\min_{\phi, \theta, \theta_c} \left\{ \mathcal{L}_{\mathrm{cls}} - \mathcal{L}_{\mathrm{ELBO}}(\text{or } \mathcal{L}_{\mathrm{dvgae}}) + \lambda_e \mathcal{L}_e \right\} + \min_{\phi} \max_{\theta_c} \mathcal{L}_{\mathrm{nwd}}, \tag{15}$$

where $\phi$ is the parameter of the graph encoder of the VGAE, $\theta$ is the parameter of the graph decoder of the VGAE, $\theta_c$ is the parameter of the classifier and $\lambda_e$ is a hyperparameter that weighs the maximum entropy loss $\mathcal{L}_e$. It is worth noting that we balance the supervised classification loss and the NWD loss equally. In this case, our model effectively learns transferable and distinct features, leading to accurate and diverse predictions in the target domain. The complete procedures of our UDA approach for graph-structured data are summarized in Algorithm 1.

## 5 Experimental Validation

We validate our algorithm using two graph classification benchmarks. Our code is provided as a supplement.

### 5.1 Experimental Setup

**Datasets** We use the IMDB&Reddit Dataset (Yanardag & Vishwanathan, 2015) and the Ego-network Dataset (Qiu et al., 2018) in our experiments. Following the previous works, we include Coreness (Batagelj & Zaversnik, 2003), Pagerank (Page et al., 1999), Eigenvector Centrality (Bonacich, 1987), Clustering Coefficient (Watts & Strogatz, 1998), and Degree/Rarity (Adamic & Adar, 2003) as the node features for both datasets.

**IMDB&Reddit Dataset** IMDB&Reddit consists of the IMDB-Binary (1000 samples) and Reddit-Binary (2000 samples) datasets, each denoting a single domain.

- **IMDB-BINARY** Each graph in this dataset represents an ego network for an actor/actress. Nodes correspond to actors/actresses. There will be an edge between two actors/actresses who appear in the same movie. A graph is generated from either romance or action movies. The task is to classify the graph into romance or action genres.

- **REDDIT-BINARY** Each graph represents an online discussion thread. Nodes correspond to users. If one of the users has responded to another's comments, then an edge exists between them. The discussion threads are drawn from four communities: AskReddit and IAmA are question/answer-based communities. Atheism and TrollXChromosomes are discussion-based communities. The binary classification task is to classify a graph into discussion-based or question/answer-based communities.

**Ego-network Dataset** Ego-network consists of data from four social network platforms, Digg, OAG, Twitter, and Weibo, each representing a domain. Each network is modeled as a graph. Each graph has 50 nodes, and nodes in the graphs represent users. Every graph has an ego user. An edge is drawn between

two nodes if a social connection occurs between two users. The definitions of social connection of these four social network platforms are different. We extract the descriptions of social connections and social actions of each social network according to Qiu et al. (2018):

- **Digg** allows users to vote for web content such as stories and news (up or down). The social connection is users' friendship, and the social action is voting for the content.

- **OAG** is generated from AMiner and Microsoft Academic Graph. The social connection is represented as the co-authorship of users, and the social action is the citation behavior.

- **Twitter** currently known as X, the social connection on Twitter represents users' friendship, and the social action is posting tweets related to the Higgs boson, a particle discovered in 2012.

- **Weibo** is a social platform similar to Twitter. The Weibo dataset includes posting logs between September 28th, 2012, and October 29th, 2012, among 1,776,950 users. The social connection is defined as users' friendship, and the social action is re-posting messages on Weibo.

All the graphs in the four domains are labeled as active or inactive, the ego user's action status. If the user makes the social action, then the user is active. The task is to identify whether the ego users are active or inactive. In addition to previously referenced node features, Ego-nework dataset also contains DeepWalk embeddings for each node (Perozzi et al., 2014), the number/ratio of active neighbors (Backstrom et al., 2006), the density of subnetwork induced by active neighbors (Ugander et al., 2012), and the number of connected components formed by active neighbors (Ugander et al., 2012).

**Baselines for Comparison**    There is a limited number of UDA algorithms specifically designed for graph classification tasks. As a result, we conduct a comparative analysis between our proposed method and updated versions of several representative methods (DANN, MDD, DIVA) and current state-of-the-art UDA methods (SDAT, BIWAA, ToAlign) for array-structured data. To facilitate the adaptation of these algorithms to graph-structured data and consistent with the structure of our VGAE's graph encoder, we replace the feature extraction backbones originally designed for array-structured data with GATs. These methods are explained below:

- **Sources:** Plain GATs trained without domain adaptation techniques.

- **DANN:** Domain Adversarial Neural Network (DANN) (Ganin et al., 2016) adopts an adversarial learning strategy. It contains a domain classifier. The domain classifier tries to distinguish the samples from which domain and the feature extractor aims to confuse the domain classifier.

- **MDD:** Margin Disparity Discrepancy (MDD) (Zhang et al., 2019b) is first proposed for computer vision tasks. It measures the distribution discrepancy and is tailored to the minimax optimization for training.

- **DIVA:** Domain Invariant Variational Autoencoders (DIVA) (Ilse et al., 2020) disentangles the inputs into three latent variables, domain latent variables, semantic latent variables, and residual variations latent variables. It is proposed to solve problems in fields such as medical imaging.

- **SDAT:** Smooth Domain Adversarial Training (SDAT) (Rangwani et al., 2022) focuses on achieving smooth minima with respect to classification loss, which stabilizes adversarial training and improves the performance on the target domain.

- **BIWAA:** Backprop Induced Feature Weighting for Adversarial Domain Adaptation with Iterative Label Distribution Alignment (BIWAA) (Westfechtel et al., 2023) employs a classifier-based backprop-induced weighting of the feature space, allowing the domain classifier to concentrate on features that are important for classification and coupling the classification and adversarial branch more closely.

- **ToAlign:** Task-oriented Alignment for Unsupervised Domain Adaptation (ToAlign) (Wei et al., 2021) decomposes features in the source domain into classification task-related and classification task-irrelevant parts under the guidance of classification meta-knowledge, ensuring that the domain adaptation is beneficial for the performance on the classification task.

**Evaluation Metrics**  Following the literature (Cai et al., 2024), the F1-Score is employed as a metric for the quantitative assessment of all methods. This score, which is the harmonic mean of precision and recall. The formula for the F1-Score is as follows:

$$F_1 = 2 \cdot \frac{\text{precision} \cdot \text{recall}}{\text{precision} + \text{recall}} \tag{16}$$

**Training Scheme**  In our evaluation, we rigorously train five models for each baseline method by employing five distinct random seeds for parameter initialization and dropping or adding edges during the data augmentation phase. We report both the average performance and standard deviation of the obtained F1-score. To ensure a fair comparison across all methods, we maintain the same seed for data shuffling. The optimization process uses the Adam (Kingma & Ba, 2014) optimizer. Please refer to the last section in the Appendix for a comprehensive description of our training scheme.

## 5.2  Performance Results and Comparison

Tables 1 and 2 present our performance results. The bold font denotes the highest performance in each column.

Table 1: Performance results on Ego-network dataset

| Method | O→T | O→W | O→D | T→O | T→W | T→D | W→O | W→T | W→D | D→O | D→T | D→W | Avg |
|---|---|---|---|---|---|---|---|---|---|---|---|---|---|
| Source | $40.0_{\pm0.0}$ | $40.4_{\pm0.3}$ | $43.8_{\pm2.4}$ | $40.2_{\pm0.0}$ | $\mathbf{48.0}_{\pm1.2}$ | $41.3_{\pm0.0}$ | $40.2_{\pm0.0}$ | $46.6_{\pm0.9}$ | $41.3_{\pm0.1}$ | $40.2_{\pm0.0}$ | $40.0_{\pm0.0}$ | $39.8_{\pm0.0}$ | 41.8 |
| DANN | $42.0_{\pm0.6}$ | $41.7_{\pm0.6}$ | $51.3_{\pm0.8}$ | $40.7_{\pm0.7}$ | $42.0_{\pm0.9}$ | $49.9_{\pm1.7}$ | $40.3_{\pm0.1}$ | $41.3_{\pm0.6}$ | $50.9_{\pm0.4}$ | $40.3_{\pm0.0}$ | $40.4_{\pm0.3}$ | $42.5_{\pm0.7}$ | 43.6 |
| MMD | $40.2_{\pm0.1}$ | $41.2_{\pm1.1}$ | $48.0_{\pm3.2}$ | $40.2_{\pm0.0}$ | $45.5_{\pm1.5}$ | $41.3_{\pm0.0}$ | $40.2_{\pm0.0}$ | $46.2_{\pm2.9}$ | $41.9_{\pm1.3}$ | $40.2_{\pm0.0}$ | $40.1_{\pm0.1}$ | $40.0_{\pm0.2}$ | 42.1 |
| DIVA | $42.1_{\pm0.5}$ | $42.4_{\pm1.4}$ | $48.9_{\pm0.7}$ | $40.3_{\pm0.2}$ | $42.0_{\pm0.4}$ | $50.3_{\pm0.5}$ | $40.4_{\pm0.3}$ | $41.2_{\pm0.3}$ | $48.7_{\pm0.9}$ | $\mathbf{40.6}_{\pm0.5}$ | $41.3_{\pm0.4}$ | $42.5_{\pm0.5}$ | 43.4 |
| SDAT | $40.2_{\pm0.1}$ | $40.1_{\pm0.5}$ | $42.2_{\pm1.6}$ | $40.2_{\pm0.0}$ | $41.6_{\pm1.3}$ | $42.9_{\pm3.2}$ | $40.3_{\pm0.2}$ | $41.1_{\pm0.7}$ | $43.1_{\pm2.8}$ | $40.2_{\pm0.0}$ | $40.1_{\pm0.1}$ | $39.9_{\pm0.1}$ | 41.0 |
| BIWAA | $40.1_{\pm0.2}$ | $41.5_{\pm0.2}$ | $45.1_{\pm1.3}$ | $40.3_{\pm0.2}$ | $\mathbf{48.0}_{\pm2.9}$ | $43.3_{\pm3.0}$ | $40.2_{\pm0.0}$ | $\mathbf{50.4}_{\pm0.5}$ | $45.7_{\pm3.8}$ | $40.3_{\pm0.1}$ | $41.0_{\pm0.9}$ | $\mathbf{43.1}_{\pm0.3}$ | 43.2 |
| ToAlign | $36.5_{\pm13.3}$ | $43.0_{\pm0.9}$ | $49.1_{\pm1.4}$ | $\mathbf{40.8}_{\pm0.3}$ | $43.1_{\pm1.1}$ | $50.2_{\pm0.5}$ | $40.7_{\pm0.4}$ | $42.8_{\pm1.3}$ | $48.4_{\pm3.2}$ | $40.5_{\pm0.2}$ | $43.4_{\pm0.9}$ | $42.6_{\pm1.6}$ | 43.4 |
| DNAN | $\mathbf{42.9}_{\pm1.6}$ | $\mathbf{43.4}_{\pm1.2}$ | $\mathbf{53.7}_{\pm0.6}$ | $\mathbf{40.8}_{\pm0.2}$ | $45.3_{\pm3.0}$ | $\mathbf{53.9}_{\pm1.0}$ | $\mathbf{40.8}_{\pm0.4}$ | $48.6_{\pm0.8}$ | $\mathbf{53.4}_{\pm2.9}$ | $\mathbf{40.6}_{\pm0.2}$ | $\mathbf{44.1}_{\pm1.5}$ | $42.8_{\pm0.9}$ | $\mathbf{45.9}$ |

Table 2: Perofmrance results on IMDB&Reddit dataset

| Task | Source | DANN | MMD | DIVA | SDAT | BIWAA | ToAlign | DNAN |
|---|---|---|---|---|---|---|---|---|
| I→R | $63.4_{\pm0.2}$ | $63.9_{\pm0.8}$ | $63.7_{\pm0.4}$ | $63.6_{\pm0.5}$ | $63.6_{\pm0.6}$ | $64.0_{\pm0.8}$ | $63.3_{\pm0.2}$ | $\mathbf{64.2}_{\pm0.6}$ |
| R→I | $72.3_{\pm1.7}$ | $72.0_{\pm1.7}$ | $73.6_{\pm1.7}$ | $71.1_{\pm0.3}$ | $74.1_{\pm2.0}$ | $71.4_{\pm1.0}$ | $73.4_{\pm0.8}$ | $\mathbf{74.9}_{\pm2.0}$ |
| Avg | 67.8 | 68.0 | 67.3 | 68.0 | 68.8 | 67.7 | 68.3 | $\mathbf{69.6}$ |

**Ego-network Results**  Results for this dataset are presented in Table 1. In this benchmark, twelve UDA tasks can be defined by pairing the four domains. Our experimental results indicate that the DNAN performs the best on average and achieves state-of-the-art performance on nine tasks: O to T, O to W, O to D, T to O, T to D, W to O, W to D, D to O, and D to T. DNAN has good performances on T to W and W to T, and achieve SOTA performance on D to T and T to D tasks, showing that DNAN can successfully handle similar domains, as Digg, Twitter, and Weibo are similar content-sharing platforms. Notably, it exceeds the second-best methods by about 4% on T to D and about 3% on W to D. In addition, DNAN can also achieve SOTA performance when there is a large distribution gap between domains, such as on tasks between OAG and Twitter or OAG and Weibo. It is important to underscore that no single method can achieve the best performance on all tasks, likely due to the diverse range of domain gaps.

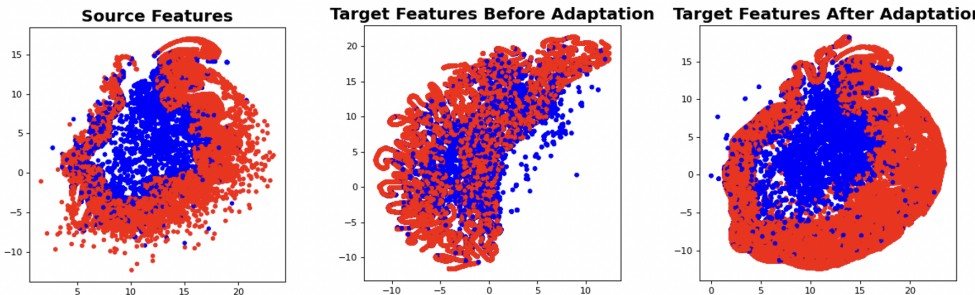

Figure 2: UMAP visualizations showing the test data representations before softmax activation on the Oag to Weibo task. Blue and red points denote the different classes. The middle plot displays the target domain data representations obtained from a model trained on the source dataset before adaptation. The left and right plots show the source and target domain data representations after adaptation using DNAN.

**IMDB&Reddit Results**   Results for this dataset are presented in Table 2. We note that the experiments demonstrate that all UDA methods perform better on the Reddit to IMDB task (R to I) than on the IMDB to Reddit task (I to R), indicating that the two tasks are not equally challenging. We hypothesize that the smaller size of IMDB-Binary compared to Reddit-Binary may result in performance degradation when testing on the larger Reddit-Binary dataset, as less knowledge can be transferred to the target domain. Our experiment results show that DNAN outperforms all other methods on both the R to I and I to R tasks, leading to SOTA results on average. On the I to R task, we observe that all methods perform similarly. Though DNAN does not outperform other methods by a large margin on the I to R task, the results still indicate that DNAN has a competitive performance compared to other methods. It is worth noting that the performance of a UDA algorithm may vary to some extent based on hyperparameter tuning. Therefore, when comparing two UDA algorithms with similar performance, they should be considered equally competitive. Based on this consideration, we can conclude that our proposed method performs competitively on all the UDA tasks and outperforms other UDA methods on average. These findings suggest that DNAN can serve as a robust UDA algorithm.

### 5.3   Analytic and Ablative and Experiments

We first perform analytic experiments to offer a deeper insight into our approach. We then performed an ablative experiment to demonstrate that all components in our algorithm are important to achieve optimal performance.

**The effect of DNAN on data representations in the output space of the classifier**   To evaluate the effectiveness of our proposed approach, we analyze how DNAN influences the target domain's distribution in the classifier's output space on the Oag to Weibo task (O to W). We picked this task to demonstrate the effect of our model because it is challenging as Weibo and Oag are very dissimilar platforms; one is connected by co-authorship, and the other is friendship. We utilize the UMAP (McInnes et al., 2018) visualization tool and compare the representations of the source domain's test data, the target domain's test data before using DNAN, and the target domain's test data after applying DNAN. In Figure 2, each point represents a single data point in the output space of the classifier before the softmax activation. Blue and red colors denote the two classes. In Figure 2, the middle plot shows that the classifier doesn't work well with the target domain data before adaptation. It's hard to distinguish the class boundary as red dots are mixed with blue ones. However, after applying DNAN, the class boundary becomes clearer, and the data representation distribution of the target domain matches well with the source domain. This is evident in the left and right plots of Figure 2, where the patterns of dots are consistent. These visualization results demonstrate that DNAN successfully mitigates the performance degradation caused by the domain shift.

**Ablative study** The ablation experiments are conducted to demonstrate the effectiveness of the two ideas we benefit from to develop DNAN. To this end, we remove one of the two components at a time and report

Table 3: Ablation Study Results on Ego-network Dataset

| Method | O→T | O→W | O→D | T→O | T→W | T→D | W→O | W→T | W→D | D→O | D→T | D→W | Avg |
|---|---|---|---|---|---|---|---|---|---|---|---|---|---|
| DNAN-D | $42.8_{\pm1.3}$ | $42.5_{\pm1.7}$ | $52.3_{\pm2.6}$ | $40.5_{\pm0.2}$ | $\mathbf{46.9}_{\pm2.0}$ | $50.0_{\pm3.5}$ | $40.4_{\pm0.2}$ | $\mathbf{50.1}_{\pm0.6}$ | $52.8_{\pm2.4}$ | $\mathbf{40.6}_{\pm0.2}$ | $42.1_{\pm1.3}$ | $42.5_{\pm1.8}$ | $45.4$ |
| DNAN-N | $\mathbf{44.4}_{\pm2.2}$ | $43.1_{\pm0.7}$ | $52.6_{\pm1.0}$ | $\mathbf{41.0}_{\pm0.3}$ | $45.0_{\pm2.4}$ | $52.7_{\pm2.7}$ | $40.7_{\pm0.3}$ | $46.9_{\pm2.2}$ | $53.3_{\pm2.5}$ | $40.5_{\pm0.2}$ | $43.0_{\pm1.1}$ | $\mathbf{43.6}_{\pm0.9}$ | $45.6$ |
| DNAN-L | $44.1_{\pm1.2}$ | $43.1_{\pm1.2}$ | $53.6_{\pm1.4}$ | $40.8_{\pm0.3}$ | $46.6_{\pm2.7}$ | $52.9_{\pm3.4}$ | $\mathbf{40.8}_{\pm0.3}$ | $48.0_{\pm2.6}$ | $53.3_{\pm3.3}$ | $40.5_{\pm0.2}$ | $43.0_{\pm0.6}$ | $\mathbf{44.3}_{\pm1.3}$ | $\mathbf{45.9}$ |
| DNAN | $42.9_{\pm1.6}$ | $\mathbf{43.4}_{\pm1.2}$ | $\mathbf{53.7}_{\pm0.6}$ | $40.8_{\pm0.2}$ | $45.3_{\pm3.0}$ | $\mathbf{53.9}_{\pm1.0}$ | $\mathbf{40.8}_{\pm0.4}$ | $48.6_{\pm0.8}$ | $\mathbf{53.4}_{\pm2.9}$ | $\mathbf{40.6}_{\pm0.2}$ | $44.1_{\pm1.5}$ | $42.8_{\pm0.9}$ | $\mathbf{45.9}$ |

Table 4: Ablation Study Results on IMDB&Reddit Dataset

| Task | DNAN-D | DNAN-N | DNAN-L | DNAN |
|---|---|---|---|---|
| I to R | $63.8_{\pm0.4}$ | $64.0_{\pm0.5}$ | $\mathbf{64.2}_{\pm0.4}$ | $\mathbf{64.2}_{\pm0.6}$ |
| R to I | $72.3_{\pm2.4}$ | $74.2_{\pm1.8}$ | $73.9_{\pm2.0}$ | $\mathbf{74.9}_{\pm2.0}$ |
| Avg | $68.0$ | $69.0$ | $69.1$ | $\mathbf{69.6}$ |

our performance. We denote the ablated versions of DNAN as: (i) **DNAN-D**: We exclude the denoising mechanism and only apply the NWD loss and the maximum entropy loss. (ii) **DNAN-N**: We exclude the NWD loss and only apply the denoising mechanism and the maximum entropy loss. (iii) **DNAN-L**: We exclude the maximum entropy loss and only apply the NWD loss and the denoising mechanism.

Our ablation study results for the Ego-network and the IMDB&Reddit datasets are illustrated in Table 3 and 4, respectively. The Ego-network dataset results reveal that the integration of both NWD loss and the denoising mechanism (DNAN) yields the highest average performance at 45.9%. The DNAN-D configuration, lacking the denoising mechanism, shows competitive performance with an average of 45.4%. However, the DNAN-N configuration, which excludes NWD loss, displays an even smaller decrease in performance, with an average of 45.6%. For the IMDB&Reddit dataset, the full DNAN model again demonstrates superior performance with an average score of 69.6%. Interestingly, the DNAN-N variant outperforms DNAN-D with averages of 69.0% and 68.0%, respectively. This observation indicates that the denoising mechanism is more critical in this context. The results from DNAN-L indicate that the entropy term does not play as important a role as the methods we have introduced. While the average performance on the Ego-network Dataset remains unchanged, DNAN-L falls short of SOTA results in as many tasks as DNAN. Moreover, there's a reduction, specifically by 0.5%, in the average performance on the IMDB&Reddit Dataset. The maximum entropy can be viewed as a auxiliary method that provides support. The ablation study highlights the importance of both the denoising mechanism and NWD loss in our proposed method, and confirms the . While the NWD loss and the denoising techniques contribute more evenly to the Ego-network dataset, the denoising mechanism is more beneficial for the IMDB&Reddit dataset. This suggests that the effectiveness of each component is context-dependent.

Additionally, we have performed an analysis of hyperparameter sensitivity and time and model complexity analysis. These findings are presented in Sections C and D of the Appendix, respectively.

## 6  Conclusions

We developed a new UDA method, which is specifically designed for graph-structured data. Our proposed method includes denoising and using the NWD for domain alignment in a shared embedding space. The experiments demonstrate our approach to be a promising method. By innovatively combining domain alignment through NWD with a denoising mechanism via a Variational Graph Autoencoder, DNAN has outperformed state-of-the-art methods across two major benchmarks without adding significant computational overload. The ability of our method to handle subtle and significant domain differences showcases its versatility and robustness. From ablative studies, the two ideas that DNAN benefits from are proven to be crucial for optimal performance. Future work can explore extending our approach to partial domain adaptation scenarios or situations where the source domain data can not be directly accessible.

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

# A  Nuclear-norm Wasserstein Discrepancy

**From Intra-class and Inter-class Correlations to Domain Discrepancy**   Consider a prediction matrix $P \in \mathbb{R}^{b \times k}$ predicted by classifier $C$, where $b$ represents the number of samples and $k$ represent the number of classes. $P$ has the following properties:

$$\sum_{j=1}^{k} P_{ij} = 1, \ P_{ij} \geq 0, \ \forall i \in 1, 2, ...b \tag{17}$$

The self-correlation matrix $R \in \mathbb{R}^{k \times k}$ then can be computed by $R = Z^T Z$. The intra-class correlation $I_a$ is defined as the sum of the main diagonal elements in $R$, and the inter-class correlation $I_e$ is defined as the sum of the off-diagonal elements in $R$:

$$I_a = \sum_{i,j=1}^{k} R_{ij}, \ I_e = \sum_{i \neq j}^{k} R_{ij} \tag{18}$$

The $I_a$ and $I_e$ are very different for source and target domains. For the source domain, the $I_a$ is large while the $I_e$ is relatively small, as we train with labels available so that most samples are correctly classified. For the target domain, the $I_a$ is small while the $I_e$ is relatively large due to the lack of supervised training. Based on linear algebra, we can represent $I_a = \|P\|_F$, the Frobenius norm of $P$, and

$$I_a - I_e = 2\|P\|_F - b \tag{19}$$

For the source domain, $I_a - I_e$ will be large; for the target domain, $I_a - I_e$ will be small. Therefore, $I_a - I_e$ can represent the discrepancy between two domains. Since the prediction matrix $P$ is generated by the classifier $C$, we can rewrite $P = C(Z)$, where $Z$ is the feature representation of the sample from either the source or the target domain. With inspiration from WGAN (Arjovsky et al., 2017) and 1-Wasserstein distance, the domain discrepancy can be formally formulated as

$$W_F(D_s, D_t) = \sup_{\|\|C\|_F\|_L \leq K} \mathbb{E}_{Z_s \sim D_s}[\|C(Z_s)\|_F] - \mathbb{E}_{Z_t \sim D_t}[\|C(Z_t)\|_F] \tag{20}$$

We call $W_F(D_s, D_t)$ the Frobenius norm-based 1-Wasserstein distance, where $D_s$ denotes the source domain, $D_t$ denotes the target domain, $\|\cdot\|_L$ denotes the Lipschitz semi-norm (Villani et al., 2009), and $K$ denotes the Lipschitz constant.

**From Frobenius Norm to Nuclear Norm**   From the domain discrepancy formulated above, we can see that the classifier $C$ works like a discriminator in GAN. Therefore, we can perform adversarial training to train the feature generator via $W_F(D_s, D_t)$. However, adversarial training with $W_F(D_s, D_t)$ limits the diversity of predictions. This is because it tends to push the samples in a class with fewer samples near the decision boundary closer to a neighboring class with a significantly larger number of samples far from the decision boundary (Cui et al., 2021). To address this limitation, the author proposes to use the nuclear norm instead of the Frobenius norm. The nuclear norm has been shown to be bound by the Frobenius norm (Chen et al., 2022). In addition, maximizing the nuclear norm maximizes the rank of the prediction matrix $P$ when $\|\cdot\|_F$ is nearby $\sqrt{b}$ (Cui et al., 2020; 2021). In consequence, the diversity of predictions will be enhanced. Thus, the domain discrepancy can be improved to be

$$W_N(D_s, D_t) = \sup_{\|\|C\|_*\|_L \leq K} \mathbb{E}_{Z_s \sim D_s}[\|C(Z_s)\|_*] - \mathbb{E}_{Z_t \sim D_t}[\|C(Z_t)\|_*] \tag{21}$$

$W_N(D_s, D_t)$ is called the Nuclear-norm 1-Wasserstein discrepancy (NWD). To integrate NWD into implementation, we can approximate the empirical NWD $\bar{W}_N$ by maximizing $\mathcal{L}_{\text{nwd}}$ that is defined below

$$\mathcal{L}_{\text{nwd}} = \frac{1}{N_s} \sum_{k=1}^{N_s} \|C(Z_s^k)\|_* - \frac{1}{N_t} \sum_{k=1}^{N_t} \|C(Z_t^k)\|_*, \ \bar{W}_N(D_s, D_t) \approx \max \mathcal{L}_{\text{nwd}} \tag{22}$$

where $Z_s^k$ is the feature representation of the $k$th sample in the source domain dataset and $Z_t^k$ is the feature representation of the $k$th sample in the target domain dataset. $N_s$ and $N_t$ represent the number of samples in the source and target domain, respectively.

## B   The Denoising Variational Lower Bound for VGAE

The concept of using a denoising criterion in the context of variational autoencoders is discussed in the Denoising Variational Autoencoder (DVAE) paper (Im Im et al., 2017). We're using the similar logic from this paper to explain how the denoising mechanism can be applied to variational graph autoencoder. It helps explain why Equation 6 and Equation 8 are valid. In the following explanation, we use the corruption to refer adding noise procedure. We start with how the denoising mechanism will affect on the inference process of the variational graph autoencoder. We first translate the Proposition 1 in the DVAE paper to VGAE setting as follows.

**Proposition 1.** Let $q_\phi(Z|A^o, X) = \mathcal{N}(z|\mu_\phi(A^o, X), \sigma_\phi(A^o, X))$ be a Gaussian distribution, where $\mu_\phi(A^o, X)$ and $\sigma_\phi(A^o, X)$ are non-linear functions of $(A^o, X)$. Let $P(A^o|A)$ be a corruption distribution around $A$ and $A^o$ be the corrupt adjaceny matrix. Then,

$$\mathbb{E}_{P(A^o|A)}[q_\phi(Z|A^o, X)] = \int q_\phi(Z|A^o, X) P(A^o|A) \, dA^o \tag{23}$$

is a mixture of Gaussian.

If the distribution is over a discrete variable, the integral in Equation 23 can be replaced by a summation. It's instructive to examine the distribution in the discrete domain to understand that Equation 23 takes on the form of a Gaussian mixture. Basically, for each instance $A^o$ drawn from $P(A^o|A)$, substituting it into $q_\phi(Z|A^o, X)$ results in a Gaussian distribution. In our scenario, given that each element within the adjacency matrix can only be 0 or 1, $P(A^o|A)$ is a discrete distribution. Thus, we can formulate our case as follows.

**Example 1.** Let $A \in \{0,1\}^{N \times N}$ be the adjacency matrix of graph $G$ that has $N$ nodes, and consider a corruption distribution $P_\pi(A^o|A)$ around $A$ and $A^o$ is corrupted adjacency matrix. Then,

$$\mathbb{E}_{P_\pi(A^o|A)}[q_\phi(Z|A^o, X)] = \sum_{i=1}^{K} q_\phi(Z|A_i^o, X) P_\pi(A_i^o|A) \tag{24}$$

has the form of a finite mixture of Gaussian and the number of mixture components $K$ is $2^{N \times N}$.

From the DVAE paper (Im Im et al., 2017), the corruption process at the input can be viewed as adding a stochastic layer at the bottom of the inference network ($q_\phi$). Specifically, $P_\pi(A^o|A)$ can be seen as a neural network, with $\pi$ as its weights. $P_\pi(A^o|A)$ takes $A$ as input and outputs the corruption distribution. If the corruption distribution is explicitly defined (such as Bernoulli or Gaussian distributions), the network parameter $\pi$ can be trained through backpropagation using the reparameterization trick. It's important to highlight that an explicit formula for the corruption distribution is not a prerequisite for our analysis to be valid. This is because the Universal Approximation Theorem (Lu & Lu, 2020) states that a neural network is capable of approximating any continuous function, regardless of its explicit form, and our analysis is not involved any training process. Now, before we show the denoising variational lower bound for VGAE, we first explain the variational lower bound that includes an extra stochastic layer. We first present Lemma 0, a result we will use in the later proof. Following Lemma 0, we present the variational lower bound when an extra stochastic layer is included in Lemma 1.

**Lemma 0.** For all nonnegative measurable functions $f, g : \mathbb{R} \to [0, \infty)$ that satisfies $\int_{-\infty}^{\infty} f(X) \, dX = 1$,

$$\int_{-\infty}^{\infty} f(X) \log g(X) \, dX \le \int_{-\infty}^{\infty} f(X) \log f(X) \, dX$$

**Proof.** Let $x$ be a random variable with $f(X)$ be its probability density function. Consider the random variable $\log\left[\frac{f(X)}{g(X)}\right]$ with $\mathbb{E}_{f(X)}\left[\log\frac{g(X)}{f(X)}\right] = -\mathbb{E}_{f(X)}\left[\log\frac{f(X)}{g(X)}\right]$. By Jensen's inequality,

$$\mathbb{E}_{f(X)}\left[\log\frac{g(X)}{f(X)}\right] \leq \log\mathbb{E}_{f(X)}\left[\frac{g(X)}{f(X)}\right] = \log\left(\int_{-\infty}^{\infty} g(X)\,dX\right) = 0$$

Therefore, $\mathbb{E}_{f(X)}\left[\log g(X)\right] \leq \mathbb{E}_{f(X)}\left[\log f(X)\right]$.

**Lemma 1.** Consider an approximate posterior distribution of the following form:

$$q_\Phi(Z|A, X) = \int_{A^o} q_\varphi(Z|A^o, X)q_\psi(A^o|A)\,dA^o \tag{25}$$

Here, we use $\Phi = \{\varphi, \psi\}$. Then, given $P_\theta(A, Z) = P_\theta(A|Z)P(Z)$, we obtain the following inequality:

$$\log P_\theta(A) \geq \mathbb{E}_{q_\Phi(Z|A,X)}\left[\log\frac{P_\theta(A, Z)}{q_\varphi(Z|A^o, X)}\right] \geq \mathbb{E}_{q_\Phi(Z|A,X)}\left[\log\frac{P_\theta(A, Z)}{q_\Phi(Z|A, X)}\right] \tag{26}$$

**Proof.** By Jensen's inequality, we have

$$\mathbb{E}_{q_\Phi(Z|A,X)}\left[\log\frac{P_\theta(A, Z)}{q_\varphi(Z|A^o, X)}\right] = \mathbb{E}_{q_\psi(A^o|A)}\left[\mathbb{E}_{q_\varphi(Z|A^o, X)}\left[\log\frac{P_\theta(A, Z)}{q_\varphi(Z|A^o, X)}\right]\right]$$

$$\mathbb{E}_{q_\psi(A^o|A)}\left[\mathbb{E}_{q_\varphi(Z|A^o, X)}\left[\log\frac{P_\theta(A, Z)}{q_\varphi(Z|A^o, X)}\right]\right] \leq \log\left(\mathbb{E}_{q_\psi(A^o|A)}\left[\mathbb{E}_{q_\varphi(Z|A^o, X)}\left[\frac{P_\theta(A, Z)}{q_\varphi(Z|A^o, Z)}\right]\right]\right)$$

$$= \log\left(\mathbb{E}_{q_\psi(A^o|A)}\left[\int_Z \frac{P_\theta(A, Z)}{q_\varphi(Z|A^o, X)}q_\varphi(Z|A^o, Z)\,dZ\right]\right)$$

$$= \log\left(\int_{A^o} P_\theta(A)q_\psi(A^o|A)\,dA^o\right)$$

$$= \log\left(P_\theta(A) \cdot \int_{A^o} q_\psi(A^o|A)\,dA^o\right)$$

$$= \log P_\theta(A)$$

Therefore, the left inequality of Equation 26 holds, and now, for the right inequality,

$$\mathbb{E}_{q_\Phi(Z|A,X)}\left[\log\frac{P_\theta(A, Z)}{q_\varphi(Z|A^o, X)}\right] = \mathbb{E}_{q_\Phi(Z|A,X)}[\log P_\theta(A, Z)] - \mathbb{E}_{q_\Phi(Z|A,X)}[\log q_\varphi(Z|A^o, X)]$$

Applying Lemma 0 to the second term, we have

$$\int_Z \log q_\varphi(Z|A^o, X)q_\Phi(Z|A, X)\,dZ \leq \int_Z \log q_\Phi(Z|A, X)q_\Phi(Z|A, X)\,dZ$$

Hence,

$$\mathbb{E}_{q_\Phi(Z|A,X)}[\log q_\varphi(Z|A^o, X)] \leq \mathbb{E}_{q_\Phi(Z|A,X)}[\log q_\Phi(Z|A, X)]$$

Then, we have

$$\mathbb{E}_{q_\Phi(Z|A,X)}\left[\log\frac{P_\theta(A, Z)}{q_\varphi(Z|A^o, X)}\right] \geq \mathbb{E}_{q_\Phi(Z|A,X)}[\log P_\theta(A, Z)] - \mathbb{E}_{q_\Phi(Z|A,X)}[\log q_\Phi(Z|A, X)]$$

$$= \mathbb{E}_{q_\Phi(Z|A,X)}\left[\log\frac{P_\theta(A, Z)}{q_\Phi(Z|A, X)}\right].$$

Therefore, we obtain

$$\log P_\theta(A) \geq \mathbb{E}_{q_\Phi(Z|A,X)}\left[\log\frac{P_\theta(A, Z)}{q_\varphi(Z|A^o, X)}\right] \geq \mathbb{E}_{q_\Phi(Z|A,X)}\left[\log\frac{P_\theta(A, Z)}{q_\Phi(Z|A, X)}\right]$$

Recalling Example 1, our approximate distribution can be defined as follows.

$$\tilde{q}_\phi(Z|A, X) = \sum_{i=1}^{K} q_\phi(Z|A_i^o, X) P(A_i^o|A)$$

By treating denoising mechanism via adding one stochastic layer, now we can apply Lemma 1 and define the denosing variational lower bound as:

$$\log P_\theta(A) \geq \mathbb{E}_{\tilde{q}_\phi(Z|A,X)} \left[ \log \frac{P_\theta(A, Z)}{q_\phi(Z|A^o, X)} \right] \stackrel{\text{def}}{=} \mathcal{L}_{\text{dvgae}} \tag{27}$$

To check whether $\mathcal{L}_{\text{dvgae}}$ is a valid lower bound, we need to examine what is achieved by maximizing $\mathcal{L}_{\text{dvgae}}$. In fact, maximizing $\mathcal{L}_{\text{dvgae}}$ can minimize the expectation of KL divergence between the true posterior distribution ($P(Z|A, X)$) and approximate posterior distribution for each noised input ($q_\phi(Z|A^o, X)$). This is an effective objective as the inference network tries to map the noise-perturbed training data points to the the true posterior distribution. In Theorem 1, we prove that maximizing $\mathcal{L}_{\text{dvgae}}$ achieves the goal.

**Theorem 1.** Maximizing $\mathcal{L}_{\text{dvgae}}$ is equivalent to minimizing the following objective

$$\mathbb{E}_{P(A^o|A)} \left[ \text{KL} \left( q_\phi(Z|A^o, X) || P(Z|A, X) \right) \right] \tag{28}$$

In other words,

$$\log P_\theta(A) = \mathcal{L}_{\text{dvgae}} + \mathbb{E}_{P(A^o|A)} \left[ \text{KL} \left( q_\phi(Z|A^o, X) || P(Z|A, X) \right) \right]$$

**Proof.** Let us consider $\theta$ being fixed just for the sake of simpler analysis.

$$\begin{aligned}
\log P_\theta(A) - \mathcal{L}_{\text{dvgae}} &= \log P_\theta(A) - \mathbb{E}_{\tilde{q}_\phi(Z|A,X)} \left[ \log \frac{P_\theta(A, Z)}{q_\phi(Z|A^o, X)} \right] \\
&= \log P_\theta(A, X) - \mathbb{E}_{\tilde{q}_\phi(Z|A,X)} \left[ \log \frac{P_\theta(A, X, Z)}{q_\phi(Z|A^o, X)} \right] \\
&= \mathbb{E}_{\tilde{q}_\phi(Z|A,X)} [\log P_\theta(A, X)] - \mathbb{E}_{\tilde{q}_\phi(Z|A,X)} \left[ \log \frac{P(Z|A, X)P_\theta(A, X)}{q_\phi(Z|A^o, X)} \right] \\
&= \mathbb{E}_{\tilde{q}_\phi(Z|A,X)} \left[ \log \frac{q_\phi(Z|A^o, X)}{P(Z|A, X)} \right] \\
&= \mathbb{E}_{P(A^o|A)} \left[ \mathbb{E}_{q_\phi(Z|A^o,X)} \left[ \log \frac{q_\phi(Z|A^o, X)}{P(Z|A, X)} \right] \right] \\
&= \mathbb{E}_{P(A^o|A)} \left[ \text{KL}(q_\phi(Z|A^o, X) || P(Z|A, X)) \right]
\end{aligned}$$

Now we prove the $\mathcal{L}_{\text{dvgae}}$ is a valid variational lower bound. To train the VGAE with $\mathcal{L}_{\text{dvgae}}$, in the DVAE paper, the authors adopt the Monte Carlo sampling. In Monte Carlo sampling, we sample from the domain of the a function and we take the average of the samples to estimate the expected value of the function. The authors in DVAE apply Monte Carlo sampling twice, one to the inner expection $\mathbb{E}_{P(A^o|A)}$, one to the outer expection $\mathbb{E}_{q_\phi(Z|A,X)}$. Their approximation is shown below.

$$\mathcal{L}_{\text{dvgae}} = \mathbb{E}_{\tilde{q}_\phi(Z|A,X)} \left[ \log \frac{P_\theta(A, Z)}{q_\phi(Z|A^o, X)} \right] = \mathbb{E}_{q_\phi(Z|A^o,X)} \left[ \mathbb{E}_{P(A^o|A)} \left[ \frac{P_\theta(A, Z)}{q_\phi(Z|A^o, X)} \right] \right]$$

$$\mathcal{L}_{\text{dvgae}} = \mathbb{E}_{q_\phi(Z|A^o,X)} \left[ \mathbb{E}_{P(A^o|A)} \left[ \frac{P_\theta(A, Z)}{q_\phi(Z|A^o, X)} \right] \right] \approx \frac{1}{KM} \sum_{k=1}^{K} \sum_{m=1}^{M} \log \frac{P_\theta(A, Z^{(k|m)})}{q_\phi(Z^{(k|m)}|A_m^o, X)} \tag{29}$$

where $A_m^o \sim P(A^o|A)$ and $Z^{(k|m)} \sim q_\phi(Z|A_m^o, X)$. We also use Monte Carlo sampling in our scenario but only for the inner expectation $\mathbb{E}_{P(A^o|A)}$ since we adopt the procedure how the regular VAE is trained: (i) sample a corrupted input $(A^o, X)$, (ii) sample latent variable from $q_\phi(Z|A^o, X)$, (iii) reconstruct the original

adjacency matrix $A$. As the authors of DVAE state in their section 3.2 (Training Procedure), our procedure can be viewed as a special case for Equation 29. Our estimation is shown as follows.

$$\mathcal{L}_{\text{dvgae}} = \mathbb{E}_{q_\phi(Z|A^o, X)} \left[ \mathbb{E}_{P(A^o|A)} \left[ \frac{P_\theta(A, Z)}{q_\phi(Z|A^o, X)} \right] \right]$$

$$\approx \frac{1}{M} \sum_{m=1}^{M} \mathbb{E}_{q_\phi(Z|A_m^o, X)} \left[ \log \frac{P_\theta(A, Z)}{q_\phi(Z|A_m^o, X)} \right]$$

$$= \frac{1}{M} \sum_{m=1}^{M} \mathbb{E}_{q_\phi(Z|A_m^o, X)} \left[ \log \frac{P_\theta(A|Z)P(Z)}{q_\phi(Z|A_m^o, X)} \right]$$

$$= \frac{1}{M} \sum_{m=1}^{M} \mathbb{E}_{q_\phi(Z|A_m^o, X)} \left[ \log P_\theta(A|Z) \right] + \mathbb{E}_{q_\phi(Z|A_m^o, X)} \left[ \log \frac{P(Z)}{q_\phi(Z|A_m^o, X)} \right]$$

$$= \frac{1}{M} \sum_{m=1}^{M} \mathbb{E}_{q_\phi(Z|A_m^o, X)} \left[ \log P_\theta(A|Z) \right] - \text{KL}(q_\phi(Z|A_m^o, X)||P(Z))$$

With above reasoning, we show $\mathcal{L}_{\text{dvgae}}$ is a valid variational lower bound and Equation 8 is a valid approximation to $\mathcal{L}_{\text{dvgae}}$.

## C    Hyperparameters Sensitivity Analysis

An important concern for most algorithms is tuning the hyperparameters and measuring the performance sensitivity with respect to them. We evaluate the sensitivity of DNAN with respect to various hyperparameters on two tasks: Twitter to Digg (T to D) and Digg to Twitter (D to T). We varied the dimension of the latent embedding space, the output dimension of the graph decoder (before taking the dot products), the weight of the maximum entropy loss ($\mathcal{L}_e$), and the batch size. We present the F1-scores of the DNAN model as a linear function of these hyperparameters in Figure 3, with the blue lines representing the T to D task and the yellow lines representing D to T task. Through inspecting this figure, we deduce:

- **Dimension of the Latent Embedding Space:** We test the performance of DNAN on five different dimension sizes for the embedding space: 32, 64, 128, 256, and 300. The performance of DNAN peaks at a latent variable size of 256 for the T to D task and a less pronounced peak on the D to T task, indicating that a moderately large value for the dimension of the latent variable is beneficial for capturing the salient features of the data. Performance declines when the dimension is too small to capture the complexity or too large, potentially introducing noise or overfitting. We note, however, that the result indicates that the performance remains relatively decent for a wide range of embedding sizes.

- **Output Dimension of the Graph Decoder:** Similar to the experiments on the dimension of the latent variable, we test the performance of DNAN on five dimension sizes: 32, 64, 128, 256, and 300. The output dimension of the graph decoder shows a performance peak at 64 for the T to D task and the D to T task. This observation suggests that a moderately small representation capacity in the graph decoder is more beneficial. Compared with performances on the D to T task, the T to D task is less sensitive to this hyperparameter.

- **Weight of the $\mathcal{L}_e$:** We test the DNAN on six weights: 0.1, 0.5, 1.0, 2.0, 3.0, 5.0. The weight of the maximum entropy loss presents a clear peak at 1.0 for both the T to D and D to T tasks, suggesting that a balanced contribution of the entropy loss is critical for performance.

- **Batch Size:** We test five batch sizes: 64, 128, 256, 512, 1024. For batch size, there is a trend of increasing performance as the size grows, with a notable peak at a batch size of 1024 for the T to D and D to T tasks. This implies that the performance of DNAN benefits from larger batch sizes, possibly due to more stable gradient estimates. Compared with the T to D task, the D to T task is less affected by batch size variations.

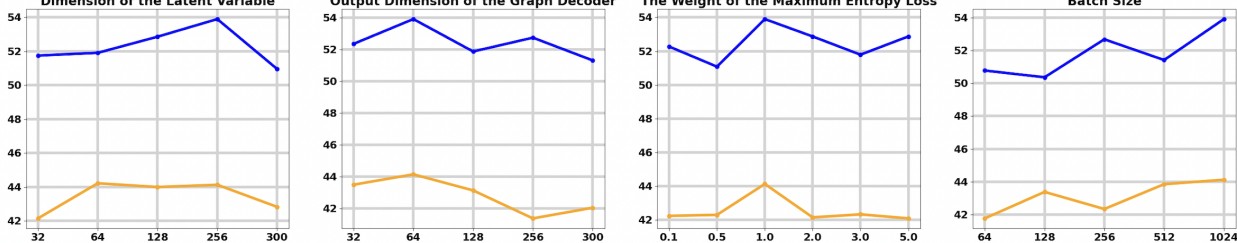

Figure 3: The performance of DNAN with different hyperparameter settings on Twitter to Digg (Blue lines) and Digg to Twitter (Yellow lines) tasks.

Table 5: Training time for IMDB&Reddit dataset

| Task | DANN | MMD | DIVA | SDAT | BIWAA | ToAlign | DNAN |
|------|------|-----|------|------|-------|---------|------|
| I→R | 10 | 260 | 14 | 12 | 899 | 10 | 10 |
| R→I | 3 | 2 | 5 | 6 | 34 | 3 | 5 |

Table 6: Model complexity on IMDB&Reddit-Binary dataset. K represents the input feature dimension, M represents the hidden dimension, and D represents the output dimension of the decoder.

| DANN | MMD | DIVA | SDAT | BIWAA | ToAlign | DNAN |
|------|-----|------|------|-------|---------|------|
| (K+3M+3)M | (K+4M+4)M | (K+11M+3D+3)M | (K+3M+4)M | (K+3M+3)M | (K+2M+11)M | (K+4M+D+2)M |

## D    Time Complexity and Model Complexity Analysis

The analysis of the training time (in minutes) and model complexity for various domain adaptation methods on the IMDB&Reddit dataset is presented in Table 5 and Table 6.

- **Training Time:** The training times reported in Table 5 illustrate the efficiency of the DNAN model relative to its counterparts. For the I to R task, DNAN required 10 minutes, positioning it the fastest in terms of training time, together with DANN and ToAlign, compared to other methods. In the R to I task, DNAN again demonstrated moderate efficiency with 5 minutes, with MMD being the fastest at 2 minutes and BIWAA the slowest at 34 minutes. These results suggest that DNAN provides a balanced trade-off between model performance and training efficiency without adding a significant computational overload.

- **Model Complexity:** The model complexity, as shown in Table 6, is assessed based on the number of parameters in the models, which is a function of the input feature dimension (K), hidden dimension (M), and the output dimension of the decoder before taking the dot products (D). Compared to other methods like DANN and SDAT, which have similar forms, DNAN introduces additional complexity due to the parameters in the graph decoder. However, it remains less complex than DIVA, which includes an extra (7M+2D+1)M term.

We conclude that the DNAN model shows competitive training time that is significantly lower than the most time-consuming method (BIWAA) while maintaining comparable or better performance. Model complexity analysis reveals that DNAN, while not the simplest, avoids the higher complexity seen in more complex methods such as DIVA. DNAN balances the computational cost with the capacity to learn and transfer knowledge effectively for better UDA performance. This observation is important because, in certain applications, it is crucial to perform UDA quickly. This is due to the constant changes in the input distribution and the limited time available to update the model. The sensitivity analysis of hyperparameters for the DNAN model on the

T to D and D to T tasks demonstrates the stability of DNAN models when using different hyperparameter values, as there is a moderate fluctuation around $\pm 3\%$. However, fine-tuning hyperparameters to the specific characteristics of the task and the dataset is beneficial. Although optimal performance is achieved with a latent variable dimension of 256, a decoder output dimension of 64, an entropy loss weight of 1.0, and a batch size of 1024, tuning the hyperparameter is not essential to achieve performance in the competitive range.

## E   Implementation Details of DNAN

In this section, we present our implementations of DNAN. Our codes are in Python, mainly with PyTorch (Paszke et al., 2019) and PyTorch Geometric (Fey & Lenssen, 2019) libraries. We train five models for every baseline using five random seeds for parameter initialization. The five random seeds are 27, 28, 29, 30, and 31. We also conducted a hyperparameter search, described in the hyperparameter sensitivity section in the paper, to find suitable hyperparameters for optimal performance. The hyperparameters we use to achieve the results listed in the main paper are presented in Table 7.

| Parameter | Ego-network | IMDB&Reddit |
| --- | --- | --- |
| Batch size | 1024 | 64 |
| Learning rate | 0.01 | 0.001 |
| Dropout rate | 0.5 | 0.2 |
| Encoder hidden size | 256 | 128 |
| Decoder output size | 64 | 128 |
| Learning decay rate | 0.75 | 0.75 |
| Entropy weight | 1.0 | 1.0 |
| Weight decay | 0.0005 | 0.0005 |
| $p_{\text{add}}$ | 0.1 | 0.1 |
| $p_{\text{drop}}$ | 0.1 | 0.1 |

Table 7: Hyper-parameters of DNAN

