# OpenReview forum: "Graph Harmony: Denoising and Nuclear-Norm Wasserstein Adaptation for Enhanced Domain Transfer in Graph-Structured Data"
_TMLR — Rejected by TMLR_

### Review · Reviewer_gVVX · 2024-02-19

**Summary Of Contributions:**

This paper proposed a new learning method for unsupervised domain adaptation (UDA) for graph domain adaptation and classification. The proposed method utilizes a graph variational autoencoder (VGAE) and a graph classifier based on a unified objective for domain adaptation, unsupervised reconstruction, classification, as well as max-entropy regularization. The method is given by eq.(10), algorithm 1, and figure 1.
The method shows comparable performance with SOTA alternatives on benchmark datasets.

**Audience:**

Yes

**Broader Impact Concerns:**

I could not identify any of such concerns.

**Claims And Evidence:**

Yes

**Requested Changes:**

In some parts the literature is mixed with the techniques, I suggest separating the literature into "related work" and introducing the technique in section 4 in a self-contained way, with pointers to the literature and related discussion focusing on the author's adaption of the literature.

sec 3. Problem description. mention whether the graphs are constrained to have the same number of nodes

sec 3. Problem description. introduce the full pipeline of domain adaptation.

sec 4. Before using the term "posterior", introduce the full generative model, only after that the term "posterior" starts to make sense.

sec 4.2. need a major revision to introduce the NWD and how it is adapted in the authors' work as compared to similar work.

sec 5. Evaluation Metrics. Need more details for the pipeline.

**Strengths And Weaknesses:**

Strength:

The proposal is well integrated with the loss, algorithm details, and comprehensive experiments.

Weaknesses:

First, the method combines several known techniques to build a unified loss function for domain adaptation (regulation, denoising, adversarial etc.) The main novelty lies in introducing nuclear-norm Wasserstein discrepancy (NWD). However, there is not a **self-contained and clear introduction to this key technique**. What is the general definition? Why can it "simultaneously achieve domain alignment and class distinguishment"?  This is missing with some external links to the literature. It is a key criterion of TMLR to have self-contained contributions.

Second, the method is **not clearly introduced and not well motivated**. The paper is organized so that the loss is introduced element by element, starting from the VGAE loss after a series of modifications. From the writing, it is not clear what is the theoretical background (justification) and why these elements are needed. For example, the author modified the ELBO in eq.(1). Then, it is natural to ask whether the modification is still a lower bound of the evidence. As another example, in section 4.2, what is the meaning of eq.(7,8) in terms of NWD?

Please see details in the "Requested Changes" section below.

---

> ### Author Response · Authors · 2024-03-03
> **Rebuttal**
>
> We thank the reviewer for the feedback. We are glad that the reviewer has found our work to be well-integrated and supported by extensive experiments. Before providing our point-by-point response to the raised concerns, we would like to highlight our motivations and contributions.
>
> **Motivations**
> - Motivations for using the denoising mechanism: samples of graphs within most application domains have a wide range of structural variations, including differences in connectivity patterns, node degrees, and subgraph structures. The primary challenge in applying UDA to graph classification is that domain shift affects structural patterns, or simply, the structural variations between the source and target domain graphs. These structural variations make it challenging for models to identify and leverage invariant features across domains. The denoising mechanism of VAGE reconstructs clean adjacency matrices from corrupted versions to address this challenge. This process forces the model to learn robust features that are more invariant to structural variations and helps the model focus on the underlying structure and features that are relevant to the classification task. Thus, the denoising mechanism helps handle the domain shift in UDA tasks for graph classification.
> - Motivation for using the Nuclear-norm Wasserstein Discrepancy (NWD): Previous GAN-based methods for UDA problems have the drawback of class mismatching, lacking clear separability between features from different classes, as they align target and source domain features irrespective of their classes. We apply NWD to address this class mismatching problem since NWD can achieve class-wise domain alignment.
>
> **Contributions: following the above motivations**
> - Our first contribution is applying denoising techniques to address the domain shift in structural patterns in the graph UDA problem. This utilization of the denoising mechanism is not trivial, and to the best of our knowledge, we're the first to apply the denoising mechanism to mitigate domain shift in the context of graph-structured data.
> - Our second contribution is integrating Nuclear-norm Wasserstein Discrepancy (NWD) with VAGE. Typically, VAGEs are used together with a domain classifier in previous domain adaptation methods. We use NWD to remove the domain classifier and incorporate the domain adaptation directly into our class classifier. This integration hasn't been done before either.

---

> ### Author Response · Authors · 2024-03-03
> **Rebuttal**
>
> **Point-by-Point Response**
>
> **First, the method combines several known techniques to build a unified loss function for domain adaptation (regulation, denoising, adversarial, etc.) The main novelty lies in introducing nuclear-norm Wasserstein discrepancy (NWD). However, there is not a self-contained and clear introduction to this key technique. What is the general definition? Why can it "simultaneously achieve domain alignment and class distinguishment"? This is missing some external links to the literature. It is a key criterion of TMLR to have self-contained contributions.**
>
> Thank you for pointing out these shortcomings. We agree that the paper should be self-contained. Due to the page limit, we provided the details of the Nuclear-norm Wasserstein in section A of the Appendix of the updated manuscript.  We included a note directing readers to this section for further information.
>
> NWD can simultaneously achieve domain alignment and class distinguishment because it addresses the class mismatch issue by incorporating class information into the domain adaptation process. The class classifier not only performs class classification but also serves as a domain discriminator. The class classifier is capable of identifying correlations both within and among different classes. These correlations, however, vary between the source domain data and the target domain data. To achieve domain alignment, NWD aligns the correlations within and between classes in the target domain with those in the source domain. This alignment ensures consistency of class across different domains, so the class mismatch problem is mitigated.
>
> **Second, the method is not clearly introduced and not well motivated. The paper is organized so that the loss is introduced element by element, starting from the VGAE loss after a series of modifications. From the writing, it is not clear what is the theoretical background (justification) and why these elements are needed. For example, the author modified the ELBO in eq.(1). Then, it is natural to ask whether the modification is still a lower bound of the evidence. As another example, in section 4.2, what is the meaning of eq.(7,8) in terms of NWD?**
>
> We explained our motivations and contributions at the beginning of the response. We hope that our response has addressed your concerns.
>
> Yes. Our modified objective function (Equation 8 in the revised manuscript) is valid. Our modified bound is an approximation of the variational lower bound under the denoising criterion (Equation 6 in the revised manuscript). The actual variational lower bound under the denoising criterion is not directly computable. The proof supporting the denoising variational lower bound and our approximation method is in section B of the Appendix in the newly uploaded manuscript.
>
> For Equations 7 and 8 in the previous manuscript, a comprehensive explanation of NWD is provided in Section A of the Appendix for understanding these equations. We have included a note directing readers to this section for further information.
>
>
> **Requested Changes**
>
> Note: We uploaded a revised manuscript and marked the changes in blue.
>
> **In some parts the literature is mixed with the techniques, I suggest separating the literature into "related work" and introducing the technique in section 4 in a self-contained way, with pointers to the literature and related discussion focusing on the author's adaptation of the literature.**
>
> Thank you for the suggestion. We modified the manuscript as you suggested. Please check section 4.1, section 4.2, and section A of the Appendix.
>
> **sec 3. Problem description. mention whether the graphs are constrained to have the same number of nodes**
>
> Thank you for highlighting this issue. Our algorithm is designed for graphs without requiring them to have an identical number of nodes. We revised it as suggested. Please check section 3, Graph Classification.
>
> **sec 3. Problem description. introduce the full pipeline of domain adaptation.**
>
> We revised it as you suggested. Please see section 3, UDA for graph classification.
>
> **sec 4. Before using the term "posterior", introduce the full generative model, only after that the term "posterior" starts to make sense.**
>
> We add the explanation about the variational inference and VAE. Please check section 4.1.
>
> **sec 4.2. need a major revision to introduce the NWD and how it is adapted in the authors' work as compared to similar work.**
>
> In our contributions section, we highlighted our adaptation of NWD into a VAGE, an effort not previously undertaken. The detailed background of NWD is presented in section A of the Appendix.
>
> **sec 5. Evaluation Metrics. Need more details for the pipeline.**
>
> We added more details for the evaluation metrics. Please see section 5, evaluation metric.

---

> ### Author Response · Authors · 2024-03-20
> **Follow-Up with the Reviewer**
>
> Dear Reviewer,
>
> We reiterate our appreciation for your time. We think that your concerns can be addressed and respectfully ask you to read our response and check the changes we did in the draft. If possible, we respectfully ask you to engage in discussion with us if you feel your concerns have not been addressed. We are hopeful that your time allows continual discussion with us so you can make your final recommendation when all your concerns are addressed.
>
> Best,
>
> Our team

---

> ### Author Response · Authors · 2024-03-27
> **Follow-UP**
>
> Dear Reviewer,
>
> We reiterate our appreciation for your time. We are hopeful to address your concerns through continual engagement. We respectfully ask you to read our response and if possible engage in discussion with us if you feel your concerns have not been addressed.
>
> Best,
>
> Our team

---

### Review · Reviewer_N2qj · 2024-02-19

**Summary Of Contributions:**

This paper tackles unsupervised domain adaptation (UDA) for the graph classification task.
To accomplish this, the paper proposes to combine prior work on Variational graph autoencoders (VGAE) and nuclear-norm Wasserstein discrepancy into a single objective.
To adapt to graph-structured data, the paper proposes to switch the standard backbones with graph transformer architecture (GAT).
The paper presents results comparing their method to multiple baselines showing marginal improvement across tasks.
Finally, the paper presents ablation study on the basic components of the method and hyperparameters.

**Audience:**

Yes

**Broader Impact Concerns:**

Not applicable.

**Claims And Evidence:**

No

**Requested Changes:**

- Clearer delineation of prior work and your contributions. Other than adapting methods to graph-structured data, there does not seem to be any clear methodological contribution. And even the graph-structured data part seems to be simply replace the networks with GAT networks.

- The paper needs more careful discussion of denoising AE and similar papers. This is not a novel contribution as claimed and should be carefully positioned within the vast literature of denoising AEs.

- Other unsupported claims about what the two components do should be carefully addressed.

- More care in defining objectives particularly the nuclear norm one.

- Better justification of entropy term.

**Strengths And Weaknesses:**

**Strengths**
- Adapted UDA methods to graph-structured data.
- Marginally outperformed baseline methods.
- Presented ablation study for various components and hyperparameters.


**Weaknesses**
- "Although MMD effectively measures distributional divergence, it may not capture higher statistical moments, an area where the Wasserstein distance (WD) (Villani, 2008) excels." - I do not believe this is true if a universal kernel is used in MMD. In theory, MMD is a universal divergence estimator that is only 0 if and only if the distributions are the same.

- The reconstruction of clean graphs from noisy graphs is exactly a denoising autoencoder. This denoising objective has been well-known for many years and is also the foundation for denoising diffusion-based models. This paper lacks references to this highly related work.

- The paper makes the following unsupported claims:
  - "This purification [i.e., denoising VAE] is crucial for aligning the source and target domain distributions." - Why is it crucial? We know that denoising AEs can learn the score function of the distribution. The "intuitive" explanation is not real support for this claim.
  - "By leveraging the Nuclear-Norm Wasserstein Discrepancy, it [DNAN] tackles the class mismatch issue in existing graph-based UDA methods." - How does it address class mismatch? Distribution matching via GAN-based emthods does not have class information. Thus, how does it solve the "class mismatch" problem theoretically or empiricaly? This claim does not seem supported.
  - "The inclusion of the denoising mechanism is also crucial as it enhances feature representation for transferability." - How do you show that it improves "transferability"? Is it just empirical results? Overall, this claim seems unsupported by the empirical results.

- "new evidence lower bound loss" - Do you prove that this is a valid ELBO objective? Is it still a proper lower bound? How is it "new"? This seems to be a simple combination of VAE and denoising AE concepts.

- If $C$ is a classifier as claimed, then it should produce a vector or class. However, Eqn 7 with the nuclear norm suggests that the ouput of $C$ is a matrix. This is confusing. After looking at the referenced paper, it seems that the input to $C$ is a *batch* of samples and the output is a *batch* of probabilitistic predictions. Is that correct? If so this should be made quite explicit as it is not clear.

- The entropy term's necessity is not clear. Has an ablation study been conducted on $\lambda_e$?

- Most of the empirical results are easily within standard deviations of other methods (and sometimes with higher variance).

- This might be a better fit for a data mining conference rather than an ML conference.

---

> ### Author Response · Authors · 2024-03-03
> **Rebuttal**
>
> We appreciate your time and extensive feedback. Before providing our point-by-point response to the raised concerns, we would like to highlight our motivations and contributions.
>
> **Motivations**
> - Motivations for using the denoising mechanism: samples of graphs within most application domains have a wide range of structural variations, including differences in connectivity patterns, node degrees, and subgraph structures. The primary challenge in applying UDA to graph classification is that domain shift affects structural patterns, or simply, the structural variations between the source and target domain graphs. These structural variations make it challenging for models to identify and leverage invariant features across domains. The denoising mechanism of VAGE reconstructs clean adjacency matrices from corrupted versions to address this challenge. This process forces the model to learn robust features that are more invariant to structural variations and helps the model focus on the underlying structure and features that are relevant to the classification task. Thus, the denoising mechanism helps handle the domain shift in UDA tasks for graph classification.
> - Motivation for using the Nuclear-norm Wasserstein Discrepancy (NWD): Previous GAN-based methods for UDA problems have the drawback of class mismatching, lacking clear separability between features from different classes, as they align target and source domain features irrespective of their classes. We apply NWD to address this class mismatching problem since NWD can achieve class-wise domain alignment.
>
> **Contributions: following the above motivations**
> - Our first contribution is applying denoising techniques to address the domain shift in structural patterns in the graph UDA problem. This utilization of the denoising mechanism is not trivial, and to the best of our knowledge, we're the first to apply the denoising mechanism to mitigate domain shift in the context of graph-structured data.
> - Our second contribution is integrating Nuclear-norm Wasserstein Discrepancy (NWD) with VAGE. Typically, VAGEs are used together with a domain classifier in previous domain adaptation methods. We use NWD to remove the domain classifier and incorporate the domain adaptation directly into our class classifier. This integration hasn't been done before either.

---

> > ### Author Response · Authors · 2024-03-03
> > **Rebuttal**
> >
> > **Point-by-Point Response**
> >
> > **"Although MMD effectively measures distributional divergence, it may not capture higher statistical moments, an area where the Wasserstein distance (WD) (Villani, 2008) excels." - I do not believe this is true if a universal kernel is used in MMD. In theory, MMD is a universal divergence estimator that is only 0 if and only if the distributions are the same.**
> >
> > Thank you for pointing out our mistakes. We had a misunderstanding about the definition of MMD and upon your comment and checking the definition, we found our our mistake. We deleted this sentence and are thankful for educating us.
> >
> > **The reconstruction of clean graphs from noisy graphs is exactly a denoising autoencoder. This denoising objective has been well-known for many years and is also the foundation for denoising diffusion-based models. This paper lacks references to this highly related work.**
> >
> > Thank you for pointing out the missing reference. We added the reference of Denosing Variational Autoencoder (DVGA) in the related work as you mentioned. Please check, section 4.1, the paragraph below Equation (4).
> >
> > **This purification [i.e., denoising VAE] is crucial for aligning the source and target domain distributions." - Why is it crucial? We know that denoising AEs can learn the score function of the distribution. The "intuitive" explanation is not real support for this claim.**
> >
> > Our experiments demonstrate that the denoising mechanism is beneficial, as the model performs better with the denoising mechanism. However, we agree that beneficial is not equal to crucial and we should have been more careful about our word choices. We changed the “crucial” to “helpful” in our manuscript. Additionally, in our motivation section at the beginning of the response, we give more details on why we believe the domain shift can be mitigated with the help of the denoising mechanism.
> >
> > **"By leveraging the Nuclear-Norm Wasserstein Discrepancy, it [DNAN] tackles the class mismatch issue in existing graph-based UDA methods." - How does it address class mismatch? Distribution matching via GAN-based methods does not have class information. Thus, how does it solve the "class mismatch" problem theoretically or empirically? This claim does not seem supported.**
> >
> > Thank you for this clarifying question. The Nuclear-norm Wasserstein Discrepancy (NWD) addresses the class mismatch issue by incorporating class information into the domain adaptation process. The class classifier not only performs category classification but also serves as a domain discriminator. The class classifier is capable of identifying correlations both within and among different classes. These correlations, however, vary between the source domain data and the target domain data. To achieve domain alignment, NWD aligns the correlations within and between classes in the target domain with those in the source domain. This alignment ensures consistency of class across different domains, so the class mismatch problem is mitigated.
> >
> > Reference: "Reusing the Task-Specific Classifier as a Discriminator: Discriminator-Free Adversarial Domain Adaptation", Lin Chen, Huaian Chen, Zhixiang Wei, Xin Jin, Xiao Tan, Yi Jin, Enhong Chen; Proceedings of the IEEE/CVF Conference on Computer Vision and Pattern Recognition (CVPR), 2022, pp. 7181-7190
> >
> > **"The inclusion of the denoising mechanism is also crucial as it enhances feature representation for transferability." - How do you show that it improves "transferability"? Is it just empirical results? Overall, this claim seems unsupported by the empirical results.**
> >
> > By “Transferability”, we mean the model's base performance in the target domain compared to a model initialized with random work. In other words, the performance that is transferred directly to the target model. Thus, the higher this source-only trained model performance is in the target domain, the greater the transferability is. If a method is said to boost transferability, it implies that it can enhance the model’s performance in the target domain. According to Table 3, DNAN-D, which lacks the denoising component, underperforms in comparison to DNAN, which includes denoising. Likewise, Table 4 indicates that DANA-D's performance is 1.6% lower than that of DNAN. These results empirically demonstrate that the denoising mechanism contributes to improved transferability.

---

> ### Author Response · Authors · 2024-03-03
> **Rebuttal**
>
> **Point-by-Point Response**
>
> **"new evidence lower bound loss" - Do you prove that this is a valid ELBO objective? Is it still a proper lower bound? How is it "new"? This seems to be a simple combination of VAE and denoising AE concepts.**
>
> Thank you for your question. We should be careful about our word choice here. Instead of using “new”, we should use “modified”. Our modified objective (Equation 8 in the revised manuscript) is valid. Our modified objective is an approximation of the variational lower bound under the denoising criterion (Equation 6 in the revised manuscript). The actual variational lower bound under the denoising criterion is not directly computable. We are thankful for your comment because we focused on doing theoretical analysis and could prove that our object function is valid. The proof supporting the denoising variational bound and our approximation method is in section B of the Appendix in the newly uploaded manuscript.
>
> **If C is a classifier as claimed, then it should produce a vector or class. However, Eqn 7 with the nuclear norm suggests that the output is a matrix. This is confusing. After looking at the referenced paper, it seems that the input is a batch of samples and the output is a batch of probabilistic predictions. Is that correct? If so this should be made quite explicit as it is not clear.**
>
> Thank you for pointing out our mistakes. The input is a batch of samples and the output is a batch of probabilistic predictions. We have made changes in our manuscript in section 4.2.  Please check the lines marked in blue right below Equation (12). The inputs in our pipeline diagram (Figure 1) were also modified.

---

> ### Author Response · Authors · 2024-03-03
> **Rebuttal**
>
> **The entropy term's necessity is not clear. Has an ablation study been conducted?**
>
> Thank you for raising this question. Entropy serves as a regularizer and aims to
> exclude redundant information from the latent variables. In response to this concern, we provide a new ablation study on the Ego-network dataset and the IMDB&Reddit Dataset. The results are shown as follows (The standard deviations are in the parentheses):
> | Methods | O to T    | O to W    | O to D    | T to O    | T to W    | T to D    | W to O    | W to T    | W to D    | D to O    | D to T    | D to W    | Avg  |
> |---------|-----------|-----------|-----------|-----------|-----------|-----------|-----------|-----------|-----------|-----------|-----------|-----------|------|
> | DNAN-L  | 44.1(1.2) | 43.1(1.2) | 53.6(1.4) | 40.8(0.3) | 46.6(2.7) | 52.9(3.4) | 40.8(0.3) | 48.0(2.6) | 53.0(3.3) | 40.5(0.2) | 43.0(0.6) | 44.3(1.3) | 45.9 |
>
> | Methods | I to R    | R to I    | Avg    |
> |---------|-----------|-----------|-----------|
> | DNAN-L  | 64.2(0.4) | 73.9(2.0) | 69.1 |
>
> The results from DNAN-L indicate that the entropy term does not play as important role as the methods we have introduced. While the average performance on the Ego-network Dataset remains unchanged, DNAN-L falls short of SOTA results in as many tasks as DNAN. Moreover, there's a reduction, specifically by 0.5%, in the average performance on the IMDB&Reddit Dataset. Though the maximum entropy does not significantly impact our model's overall performance, it serves as an auxiliary method that provides marginal benefits.
>
> **Most of the empirical results are easily within standard deviations of other methods (and sometimes with higher variance).**
>
> Please note that on average, our method exceeds the performance of alternative methods. Thus, in the paper, we claim our method is a competitive alternative rather than the best among all the methods. We would like to highlight that this situation is extremely common in the UDA literature. We respectfully ask the reviewer to check 5-10 UDA papers to see that this trend can be observed.
>
> **This might be a better fit for a data mining conference rather than an ML conference.**
>
> We respectfully think that the boundary between data mining and ML venues is not sharp and many works may be suitable for both venues. Additionally, please check the following publications in TMLR which are very similar to our work in terms of relevance to data mining:
>
> https://openreview.net/pdf?id=JFaZ94tT8M
>
> https://openreview.net/pdf?id=SSqOqAwpN7
>
> https://openreview.net/forum?id=h4BYtZ79uy
>
> The above works demonstrate that TMLR also accepts paper with a focus on data mining.
>
> Finally, we would like to especially thank you for your comments because they helped us to correct several errors and improve the theoretical justification of our work.

---

> ### Author Response · Authors · 2024-03-03
> **Rebuttal**
>
> **Requested Changes**
>
> Note: We uploaded a revised manuscript and marked the changes in blue.
>
> **Clearer delineation of prior work and your contributions. Other than adapting methods to graph-structured data, there does not seem to be any clear methodological contribution. And even the graph-structured data part seems to simply replace the networks with GAT networks.**
>
> Thank you for raising these points. We clarified our contributions and motivations in the introduction section.
>
> **The paper needs a more careful discussion of denoising AE and similar papers. This is not a novel contribution as claimed and should be carefully positioned within the vast literature of denoising AEs.**
>
> We included references for the Denoising Variational Autoencoder. If there are other missing references, please let us know. Please check, section 4.1, the paragraph below Equation (4). Regarding inquiries about the novelty of our work, please refer to the response provided in the preceding question.
>
> **Other unsupported claims about what the two components do should be carefully addressed.**
>
> We updated our writing accordingly and the places we changed are mentioned in the Weakness section.
>
> **More care in defining objectives, particularly the nuclear norm one.**
>
> We have modified the explanation of the nuclear norm and corrected our mistakes.  Please check the lines right below Equation (12) and the inputs in our pipeline diagram (Figure 1) were also modified to batch.
>
> **Better justification of the entropy term.**
>
> We added the ablation study of the entropy term. Please check the results in Table 3& Table 4 and the analysis on page 12, first paragraph.

---

> ### Author Response · Authors · 2024-03-20
> **Follow-Up with the Reviewer**
>
> Dear Reviewer,
>
> We reiterate our appreciation for your time. We think that your concerns can be addressed and respectfully ask you to read our response and check the changes we did in the draft. If possible, we respectfully ask you to engage in discussion with us if you feel your concerns have not been addressed. We are hopeful that your time allows continual discussion with us so you can make your final recommendation when all your concerns are addressed.
>
> Best,
>
> Our team

---

> > ### Comment · Reviewer_N2qj · 2024-03-22
> >
> > I have read the author response. It has improved the paper and addressed some clear mistakes. I have taken the author response into consideration for my final recommendation.

---

> > > ### Author Response · Authors · 2024-03-22
> > > **Question about the remaining concerns**
> > >
> > > Dear Reviewer,
> > >
> > > Thank you for reading our response and fitting this task in your schedule. We are glad that it has addressed some of the concerns and corrected the mistakes. If there are any unaddressed ones, please let us know so we try to address all your concerns.
> > >
> > > Best,
> > >
> > > Our team

---

> > > ### Author Response · Authors · 2024-03-27
> > > **Request for Reviewer N2qj**
> > >
> > > Dear Reviewer,
> > >
> > > We are grateful that you have read our response. Irrespective of your recommendation, we appreciate you list the concerns that our response has not addressed. This will help us to improve our work for future, irrespective of the outcome for the current submission.
> > >
> > > Thank you,
> > >
> > > Our team

---

### Review · Reviewer_1Qu8 · 2024-02-20

**Summary Of Contributions:**

To address the challenge of labeled cost in graph classification tasks, this study applies unsupervised domain adaptation (UDA) techniques to utilize labeled nodes from related source domains to enhance an unlabeled target domain. Traditionally, UDA methods, particularly those based on Generative Adversarial Networks (GANs), have focused primarily on array-structured data, such as images, and not on graph-structured data. Consequently, this paper introduces a Denoising and Nuclear-Norm Wasserstein Adaptation Network (DNAN), specifically designed to overcome the limitations of GAN-based UDA approaches when applied to graph-structured data.

**Audience:**

Yes

**Broader Impact Concerns:**

Actually, there is no serious concern about the ethical implications of this work.

**Claims And Evidence:**

No

**Requested Changes:**

Same as in the weakness in the above section.

Pls:
The quality of the paper, especially within the introduction, requires refinement to ensure the main contributions and motivations are clearly and explicitly presented, rather than being cursorily addressed.

**Strengths And Weaknesses:**

Strengths:

1. Applying the unsupervised domain adaptation (UDA) method to graph classification tasks represents a potentially innovative approach.


Weakness:

1. Alignment of Motivation and Solution:

(1) Motivation Clarification: The initial motivation of this research is to mitigate the labeling cost associated with graph-structured data. However, the subsequent focus shifts predominantly to unsupervised domain adaptation (UDA) methodologies without explicitly linking back to how these methods alleviate the label scarcity issue. To enhance coherence, it's crucial to establish a direct connection between the UDA techniques employed and their impact on reducing labeling costs.

(2) Solution and Justification for Denoising: The introduction of a denoising mechanism as part of the proposed method is intriguing but lacks a clear justification. To resolve this, it is essential to define what constitutes "noise" in the context of graph-structured data within UDA settings and elaborate on how this noise adversely affects the adaptation process.

2. Clarification of Contributions:

The paper should delineate the distinctions and challenges unique to graph classification tasks, especially in UDA contexts, compared to traditional i.i.d (independently and identically distributed) image classification and non-i.i.d node classification scenarios. Highlighting the specific characteristics of i.i.d and non-i.i.d graph datasets, and the inherent difficulties in applying UDA methods to graph classification, would underscore the novelty and significance of your contributions.

3. Experimental Setup and Objectives:

The main experiments need a clearer articulation of their objectives, particularly in demonstrating the effectiveness of the proposed method in addressing label scarcity. It's essential to outline a comprehensive experimental design that explicitly tests the proposed method's ability to leverage unlabeled data effectively, thereby reducing the dependence on labeled data.

---

> ### Author Response · Authors · 2024-03-03
> **Rebuttal**
>
> We thank the reviewer for the feedback. We are glad that the reviewer has found our work to be innovative. Before providing our point-by-point response to the raised concerns, we would like to highlight our motivations and contributions.
>
> **Motivations**
> - Motivations for using the denoising mechanism: samples of graphs within most application domains have a wide range of structural variations, including differences in connectivity patterns, node degrees, and subgraph structures. The primary challenge in applying UDA to graph classification is that domain shift affects structural patterns, or simply, the structural variations between the source and target domain graphs. These structural variations make it challenging for models to identify and leverage invariant features across domains. The denoising mechanism of VAGE reconstructs clean adjacency matrices from corrupted versions to address this challenge. This process forces the model to learn robust features that are more invariant to structural variations and helps the model focus on the underlying structure and features that are relevant to the classification task. Thus, the denoising mechanism helps handle the domain shift in UDA tasks for graph classification.
> - Motivation for using the Nuclear-norm Wasserstein Discrepancy (NWD): Previous GAN-based methods for UDA problems have the drawback of class mismatching, lacking clear separability between features from different classes, as they align target and source domain features irrespective of their classes. We apply NWD to address this class mismatching problem since NWD can achieve class-wise domain alignment.
>
> **Contributions: following the above motivations**
> - Our first contribution is applying denoising techniques to address the domain shift in structural patterns in the graph UDA problem. This utilization of the denoising mechanism is not trivial, and to the best of our knowledge, we're the first to apply the denoising mechanism to mitigate domain shift in the context of graph-structured data.
> - Our second contribution is integrating Nuclear-norm Wasserstein Discrepancy (NWD) with VAGE. Typically, VAGEs are used together with a domain classifier in previous domain adaptation methods. We use NWD to remove the domain classifier and incorporate the domain adaptation directly into our class classifier. This integration hasn't been done before either.

---

> > ### Author Response · Authors · 2024-03-03
> > **Rebuttal**
> >
> > **Point-by-Point Response**
> >
> > **Motivation Clarification: The initial motivation of this research is to mitigate the labeling cost associated with graph-structured data. However, the subsequent focus shifts predominantly to unsupervised domain adaptation (UDA) methodologies without explicitly linking back to how these methods alleviate the label scarcity issue. To enhance coherence, it's crucial to establish a direct connection between the UDA techniques employed and their impact on reducing labeling costs.**
> >
> > We regret that our motivation and contributions were unclear. Part of this unclarity may be because of the page limit. Please note that the challenge of label scarcity requires Unsupervised Domain Adaptation (UDA) due to the absence of labels in the target domain. The target domain is our primary area of interest but since labeled data is not accessible, we cannot train a model using supervised learning. To overcome this challenge, we utilize a source dataset rich in labels for training that is related to the target domain. However, the inherent difference between the source and target datasets requires the application of domain adaptation strategies due to the existence of a domain gap between the source and the target domain. UDA strategies enable the effective application of models trained on the well-labeled source data to achieve high performance on the target domain, despite its lack of labels. We hope our explanation has clarified the motivation.
> >
> > **Solution and Justification for Denoising: The introduction of a denoising mechanism as part of the proposed method is intriguing but lacks a clear justification. To resolve this, it is essential to define what constitutes "noise" in the context of graph-structured data within UDA settings and elaborate on how this noise adversely affects the adaptation process.**
> >
> > Following the previous explanation, our motivation for using the denoising mechanism is provided at the beginning of our response. The “noise” in our UDA setting is defined as the structural noise for graph data. Structural noise refers to inaccuracies or anomalies in the topology of the graph. For example, edges that shouldn't exist between nodes or missing edges that should exist. Structural noise affects the adaptation process in several ways:
> > - Degraded Transferability: Noisy edges can weaken the robustness and transferability of learned representations, diminishing the effectiveness of domain adaptation strategies that rely on shared graph structures.
> > - Domain Shift Amplification: In UDA, the goal is to adapt a model trained on a source domain to perform well on a target domain, where the two domains differ in distribution. Having noise in the topology of graph samples can amplify the domain shift, making it even more challenging for adaptation algorithms to bridge the gap between the source and target domains.
> > - Feature Distortion: Noise affects the propagation of node features, leading to either overly similar (smoothed) or overly distinct (sharpened) node representations. This undermines the model's capacity to accurately leverage structural information.
> > - Risk of Overfitting/Underfitting: Noisy edges may cause models to overfit to perturbed structures, reducing generalizability, or underfit due to disrupted connectivity, impeding the learning of meaningful patterns.
> >
> > We hope you found our response convincing.

---

> > > ### Author Response · Authors · 2024-03-03
> > > **Rebuttal**
> > >
> > > **Point-by-Point Response**
> > >
> > > **The paper should delineate the distinctions and challenges unique to graph classification tasks, especially in UDA contexts, compared to traditional i.i.d (independently and identically distributed) image classification and non-i.i.d node classification scenarios. Highlighting the specific characteristics of i.i.d and non-i.i.d graph datasets, and the inherent difficulties in applying UDA methods to graph classification, would underscore the novelty and significance of your contributions.**
> > >
> > > The unique challenge to graph classification in UDA contexts is the structural variations between the source and target domain graphs. These structural variations make it challenging for models to identify and leverage invariant features across domains. We clarify and underscore our motivations and contributions at the beginning of the response.
> > >
> > > **The main experiments need a clearer articulation of their objectives, particularly in demonstrating the effectiveness of the proposed method in addressing label scarcity. It's essential to outline a comprehensive experimental design that explicitly tests the proposed method's ability to leverage unlabeled data effectively, thereby reducing the dependence on labeled data.**
> > >
> > > Please note that we evaluated our algorithm on two graph domain adaptation datasets. In this setup, we have access to labels for the source domain graphs but not for the target domain graphs. Therefore, we train our model using labeled source domain graphs and unlabeled target domain graphs. Our goal is to leverage the knowledge gained from the source domain graphs to accurately classify the target domain graphs, without the dependence on target domain labels. The primary objective of these experiments is to evaluate the model's effectiveness in classifying target domain graphs without using their labels during training.
> > >
> > > Please let us know if you think the portion of the text in the paper should be specifically changed to address these concerns.
> > >
> > > **Requested Changes**
> > >
> > > **Pls: The quality of the paper, especially within the introduction, requires refinement to ensure the main contributions and motivations are clearly and explicitly presented, rather than being cursorily addressed.**
> > >
> > > Note: We uploaded a revised manuscript and marked our changes in blue.
> > >
> > > As suggested, we have clarified our motivation and contributions in the introduction. Please see the introduction section. We are more than happy to change more if you provide us with specific guidelines.

---

> ### Author Response · Authors · 2024-03-20
> **Follow-Up with the Reviewer**
>
> Dear Reviewer,
>
> We reiterate our appreciation for your time. We think that your concerns can be addressed and respectfully ask you to read our response and check the changes we did in the draft. If possible, we respectfully ask you to engage in discussion with us if you feel your concerns have not been addressed. We are hopeful that your time allows continual discussion with us so you can make your final recommendation when all your concerns are addressed.
>
> Best,
>
> Our team

---

> ### Author Response · Authors · 2024-03-27
> **Follow-Up**
>
> Dear Reviewer,
>
> We reiterate our appreciation for your time. We are hopeful to address your concerns through continual engagement. We respectfully ask you to read our response and if possible engage in discussion with us if you feel your concerns have not been addressed.
>
> Best,
>
> Our team

---

### Comment · Action_Editor_g4Ub · 2024-02-26
**Rebuttal and rolling discussion**

Dear authors,

We have collected three reviews. The reviewers will submit their final recommendation in 2 weeks. Could you try to do rebuttal and rolling discussions within the 2 weeks? If you need any help, please let us know as soon as possible.

Best wishes,
AE

---

> ### Author Response · Authors · 2024-02-27
> **Rebuttal Period**
>
> Dear Action Editor,
>
> We aim to complete the rebuttal by the end of this week. When you mentioned two weeks, did that period begin this week or last week?
>
> Best Wishes,
> Authors

---

### Author Response · Authors · 2024-03-04
**Rebuttal is Ready**

Dear Action Editor and Reviewers,

Our rebuttal has been posted. We look forward to your response.

Many thanks!
Authors

---

### Decision · Action_Editor_g4Ub · 2024-03-29

**Recommendation:** Reject

**Comment:**

After careful consideration of the feedback from Reviewer gVVX, Reviewer N2qj, and Reviewer 1Qu8, and the authors' responses, I recommend rejecting this submission. However, this decision should not discourage the authors as it is primarily driven by the lack of a "Major Revision" option in the system.

The paper requires major restructuring to clearly highlight its conceptual and technical novelty, as pointed out by Reviewer gVVX. The technical quality and clarity, especially regarding the modified ELBO and Theorem 1, need significant improvement, as also emphasized by Reviewer gVVX.

Furthermore, the paper fails to sufficiently address prior work on diffusion models, denoising models, and graph-based diffusion models, leading to overclaims and a lack of clarity in its framing, as indicated by Reviewer N2qj. Reviewer 1Qu8 also noted that the paper lacks logical coherence and depth in evaluation.


Given the substantial revisions required to address these concerns, I believe that the paper is not suitable for publication in its current form. I encourage the authors to consider the feedback provided by Reviewer gVVX, Reviewer N2qj, and Reviewer 1Qu8, and undertake a thorough revision of their work before resubmission.

**Audience:**

The paper is likely to interest researchers and practitioners working on graph classification, domain adaptation, and related areas. However, for this interest to be fully realized, the authors must address the concerns raised by the reviewers with contribution and soundness.

**Claims And Evidence:**

The paper proposes an approach to unsupervised domain adaptation (UDA) for graph classification tasks by utilizing labeled nodes from related source domains to enhance an unlabeled target domain. The approach involves the integration of Variational Graph Autoencoders (VGAE), nuclear-norm Wasserstein discrepancy, and a Graph Transformer Architecture (GAT). Some important claims are not supported well, as highlighted by different Reviewers.

**Resubmission Of Major Revision:**

The authors may consider submitting a major revision at a later time.